# Settling the Variance of Multi-Agent Policy Gradients

**Jakub Grudzien Kuba**[*,1,2], **Muning Wen**[*,3], **Linghui Meng**[4], **Shangding Gu**[4],
**Haifeng Zhang**[4], **David Henry Mguni**[2], **Jun Wang**[5], **Yaodong Yang**[†,6]
[1]Imperial College London, [2]Huawei R&D UK, [3]Shanghai Jiao Tong University,
[4]Institute of Automation, Chinese Academy of Science,
[5]University College London, [6]Institute for AI, Peking University.

## Abstract

Policy gradient (PG) methods are popular reinforcement learning (RL) methods
where a baseline is often applied to reduce the variance of gradient estimates. In
multi-agent RL (MARL), although the PG theorem can be naturally extended, the
effectiveness of multi-agent PG (MAPG) methods degrades as the variance of gra-
dient estimates increases rapidly with the number of agents. In this paper , we offer
a rigorous analysis of MAPG methods by, firstly, quantifying the contributions of
the number of agents and agents' explorations to the variance of MAPG estimators.
Based on this analysis, we derive the optimal baseline (OB) that achieves the mini-
mal variance. In comparison to the OB, we measure the excess variance of existing
MARL algorithms such as vanilla MAPG and COMA. Considering using deep neu-
ral networks, we also propose a surrogate version of OB, which can be seamlessly
plugged into any existing PG methods in MARL. On benchmarks of Multi-Agent
MuJoCo and StarCraft challenges, our OB technique effectively stabilises training
and improves the performance of multi-agent PPO and COMA algorithms by a
significant margin. Code is released at https://github.com/morning9393/
Optimal-Baseline-for-Multi-agent-Policy-Gradients.

## 1 Introduction

Policy gradient (PG) methods refer to the category of reinforcement learning (RL) algorithms where
the parameters of a stochastic policy are optimised with respect to the expected reward through
gradient ascent. Since the earliest embodiment of REINFORCE [40], PG methods, empowered by
deep neural networks [33], are among the most effective model-free RL algorithms on various kinds
of tasks [2, 6, 10]. However, the performance of PG methods is greatly affected by the variance of
the PG estimator [35, 37]. Since the RL agent behaves in an unknown stochastic environment, which
is often considered as a black box, the randomness of expected reward can easily become very large
with increasing sizes of the state and action spaces; this renders PG estimators with high variance,
which concequently leads to low sample efficiency and unsuccessful trainings [9, 35].

To address the large variance issue and improve the PG estimation, different variance reduction
methods were developed [9, 23, 35, 48]. One of the most successfully applied and extensively
studied methods is the control variate subtraction [8, 9, 37], also known as the *baseline* trick. A
baseline is a scalar random variable, which can be subtracted from the state-action value samples in
the PG estimates so as to decrease the variance, meanwhile introducing no bias to its expectation.
Baselines can be implemented through a constant value [8, 12] or a value that is dependent on the
state [8, 9, 11, 33] such as the state value function, which results in the zero-centered advantage
function and the advantage actor-critic algorithm [18]. State-action dependent baselines can also be
applied [9, 41], although they are reported to have no advantages over state-dependent baselines [35].

---

[*]Equal contribution. [†]Corresponding author <yaodong.yang@pku.edu.cn>.

35th Conference on Neural Information Processing Systems (NeurIPS 2021).

When it comes to multi-agent reinforcement learning (MARL) [43], PG methods can naturally be applied. One naive approach is to make each agent disregard its opponents and model them as part of the environment. In such a setting, single-agent PG methods can be applied in a fully decentralised way (hereafter referred as *decentralised training* (DT)). Although DT has numerous drawbacks, for example the non-stationarity issue and poor convergence guarantees [3, 16], it demonstrates excellent empirical performance in certain tasks [19, 26]. A more rigorous treatment is to extend the PG theorem to the multi-agent policy gradient (MAPG) theorem, which induces a learning paradigm known as *centralised training with decentralised execution* (CTDE) [7, 42, 45, 46]. In CTDE, each agent during training maintains a centralised critic which takes joint information as input, for example COMA [7] takes joint state-action pair; meanwhile, it learns a decentralised policy that only depends on the local state, for execution. Learning the centralised critic helps address the non-stationarity issue encountered in DT [7, 16]; this makes CTDE an effective framework for implementing MAPG and successful applications have been achieved in many real-world tasks [13, 17, 22, 38, 39, 50].

Unfortunately, compared to single-agent PG methods, MAPG methods suffer more from the large variance issue. This is because in multi-agent settings, the randomness comes not only from each agent's own interactions with the environment but also other agents' explorations. In other words, an agent would not be able to tell if an improved outcome is due to its own behaviour change or other agents' actions. Such a *credit assignment* problem [7, 36] is believed to be one of the main reasons behind the large variance of CTDE methods [16]; yet, despite the intuition being built, there is still a lack of mathematical treatment for understanding the contributing factors to the variance of MAPG estimators. As a result, addressing the large variance issue in MARL is still challenging. One relevant baseline trick in MARL is the application of a *counterfactual baseline*, introduced in COMA [7]; however, COMA still suffers from the large variance issue empirically [19].

In this work, we analyse the variance of MAPG estimates mathematically. Specifically, we try to quantify the contributions of the number of agents and the effect of multi-agent explorations to the variance of MAPG estimators. One natural outcome of our analysis is the optimal baseline (OB), which achieves the minimal variance for MAPG estimators. Our OB technique can be seamlessly plugged into any existing MAPG methods. We incorporate it in COMA [7] and a multi-agent version of PPO [29], and demonstrate its effectiveness by evaluating the resulting algorithms against the state-of-the-art algorithms. Our main contributions are summarised as follows:

1. We rigorously quantify the excess variance of the CTDE MAPG estimator to that of the DT one and prove that the order of such excess depends linearly on the number of agents, and quadratically on agents' exploration terms (i.e., the local advantages).

2. We demonstrate that the counterfactual baseline of COMA reduces the noise induced by other agents, but COMA still faces the large variance due to agent's own exploration.

3. We derive that there exists an optimal baseline (OB), which minimises the variance of an MAPG estimator, and introduce a surrogate version of OB that can be easily implemented in any MAPG algorithms with deep neural networks.

4. We show by experiments that OB can effectively decrease the variance of MAPG estimates in COMA and multi-agent PPO, stabilise and accelerate training in StarCraftII and Multi-Agent MuJoCo environments.

## 2 Preliminaries & Background

In this section, we provide the preliminaries for the MAPG methods in MARL. We introduce notations and problem formulations in Section 2.1, present the MAPG theorem in Section 2.2, and finally, review two existing MAPG methods that our OB can be applied to in Section 2.3.

### 2.1 Multi-Agent Reinforcement Learning Problem

We formulate a MARL problem as a Markov game [14], represented as a tuple $\langle \mathcal{N}, \mathcal{S}, \boldsymbol{\mathcal{A}}, \mathcal{P}, r, \gamma \rangle$. Here, $\mathcal{N} = \{1, \ldots, n\}$ is the set of agents, $\mathcal{S}$ is the state space, $\boldsymbol{\mathcal{A}} = \prod_{i=1}^{n} \mathcal{A}^i$ is the product of action spaces of the $n$ agents, known as the joint action space, $\mathcal{P} : \mathcal{S} \times \boldsymbol{\mathcal{A}} \times \mathcal{S} \to \mathbb{R}$ is the transition probability kernel, $r : \mathcal{S} \times \boldsymbol{\mathcal{A}} \to \mathbb{R}$ is the reward function (with $|r(s, \boldsymbol{a})| \leq \beta, \forall s \in \mathcal{S}, \boldsymbol{a} \in \boldsymbol{\mathcal{A}}$), and $\gamma \in [0, 1)$ is the discount factor. Each agent $i$ possesses a parameter vector $\theta^i$, which concatenated with parameters of other agents, gives the joint parameter vector $\boldsymbol{\theta}$. At a time step $t \in \mathbb{N}$, the

agents are at state $s_t \in \mathcal{S}$. An agent $i$ takes an action $a_t^i \in \mathcal{A}^i$ drawn from its stochastic policy $\pi_{\boldsymbol{\theta}}^i(\cdot|s_t)$ parametrised by $\theta^i$ [(1)], simultaneously with other agents, which together gives a joint action $\mathbf{a}_t = (a_t^1, \ldots, a_t^n) \in \mathcal{A}$, drawn from the joint policy $\boldsymbol{\pi}_{\boldsymbol{\theta}}(\mathbf{a}_t|s_t) = \prod_{i=1}^n \pi_{\boldsymbol{\theta}}^i(a_t^i|s_t)$. The system moves to state $s_{t+1} \in \mathcal{S}$ with probability mass/density $\mathcal{P}(s_{t+1}|s_t, \mathbf{a}_t)$. A trajectory is a sequence $\tau = \langle s_t, \mathbf{a}_t, r_t \rangle_{t=0}^\infty$ of states visited, actions taken, and rewards received by the agents in an interaction with the environment. The joint policy $\boldsymbol{\pi}_{\boldsymbol{\theta}}$, the transition kernel $\mathcal{P}$, and the initial state distribution $d^0$, induce the marginal state distributions at time $t$, i.e., $d_{\boldsymbol{\theta}}^t(s)$, which is a probability mass when $\mathcal{S}$ is discrete, and a density function when $\mathcal{S}$ is continuous. The total reward at time $t \in \mathbb{N}$ is defined as $R_t \triangleq \sum_{k=0}^\infty \gamma^k r_{t+k}$. The state value function $V_{\boldsymbol{\theta}}$ and the state-action value function $Q_{\boldsymbol{\theta}}$ are given by

$$V_{\boldsymbol{\theta}}(s) \triangleq \mathbb{E}_{\mathbf{a}_{0:\infty} \sim \boldsymbol{\pi}_{\boldsymbol{\theta}}, s_{1:\infty} \sim \mathcal{P}} \left[ R_0 \,\middle|\, s_0 = s \right] , \quad Q_{\boldsymbol{\theta}}(s, \boldsymbol{a}) \triangleq \mathbb{E}_{s_{1:\infty} \sim \mathcal{P}, \mathbf{a}_{1:\infty} \sim \boldsymbol{\pi}_{\boldsymbol{\theta}}} \left[ R_0 \,\middle|\, s_0 = s, \ \mathbf{a}_0 = \mathbf{a} \right] .$$

The advantage function is defined as $A_{\boldsymbol{\theta}}(s, \boldsymbol{a}) \triangleq Q_{\boldsymbol{\theta}}(s, \boldsymbol{a}) - V_{\boldsymbol{\theta}}(s)$. The goal of the agents is to maximise the expected total reward $\mathcal{J}(\boldsymbol{\theta}) \triangleq \mathbb{E}_{s \sim d^0} \left[ V_{\boldsymbol{\theta}}(s) \right]$.

In this paper, we write $-(i_1, \ldots, i_k)$ to denote the set of all agents excluding $i_1, \ldots, i_k$ (we drop the bracket when $k = 1$). We define the multi-agent state-action value function for agents $i_1, \ldots, i_k$ as $Q_{\boldsymbol{\theta}}^{i_1, \ldots, i_k} \left( s, \boldsymbol{a}^{(i_1, \ldots, i_k)} \right) \triangleq \mathbb{E}_{\mathbf{a}^{-(i_1, \ldots, i_k)} \sim \boldsymbol{\pi}_{\boldsymbol{\theta}}^{-(i_1, \ldots, i_k)}} \left[ Q_{\boldsymbol{\theta}} \left( s, \boldsymbol{a}^{(i_1, \ldots, i_k)}, \mathbf{a}^{-(i_1, \ldots, i_k)} \right) \right]$[(2)], which is the expected total reward once agents $i_1, \ldots, i_k$ have taken their actions. Note that for $k = 0$, this becomes the state value function, and for $k = n$, this is the usual state-action value function. As such, we can define the multi-agent advantage function as

$$A_{\boldsymbol{\theta}}^{i_1, \ldots, i_k} \left( s, \boldsymbol{a}^{(j_1, \ldots, j_m)}, \boldsymbol{a}^{(i_1, \ldots, i_k)} \right)$$
$$\triangleq Q_{\boldsymbol{\theta}}^{j_1, \ldots, j_m, i_1, \ldots, i_k} \left( s, \boldsymbol{a}^{(j_1, \ldots, j_m, i_1, \ldots, i_k)} \right) - Q_{\boldsymbol{\theta}}^{j_1, \ldots, j_m} \left( s, \boldsymbol{a}^{(j_1, \ldots, j_m)} \right), \tag{1}$$

which is the advantage of agents $i_1, \ldots, i_k$ playing $a^{i_1}, \ldots, a^{i_k}$, given $a^{j_1}, \ldots, a^{j_m}$. When $m = n-1$ and $k = 1$, this is often referred to as the *local advantage* of agent $i$ [7].

## 2.2 The Multi-Agent Policy Gradient Theorem

The Multi-Agent Policy Gradient Theorem [7, 47] is an extension of the Policy Gradient Theorem [33] from RL to MARL, and provides the gradient of $\mathcal{J}(\boldsymbol{\theta})$ with respect to agent $i$'s parameter, $\theta^i$, as

$$\nabla_{\theta^i} \mathcal{J}(\boldsymbol{\theta}) = \mathbb{E}_{s_{0:\infty} \sim d_{\boldsymbol{\theta}}^{0:\infty}, \mathbf{a}_{0:\infty}^{-i} \sim \boldsymbol{\pi}_{\boldsymbol{\theta}}^{-i}, a_{0:\infty}^i \sim \pi_{\boldsymbol{\theta}}^i} \left[ \sum_{t=0}^\infty \gamma^t Q_{\boldsymbol{\theta}}(s_t, \mathbf{a}_t^{-i}, a_t^i) \nabla_{\theta^i} \log \pi_{\boldsymbol{\theta}}^i(a_t^i|s_t) \right]$$

From this theorem we can derive two MAPG estimators, one for CTDE (i.e., $\mathbf{g}_C^i$), where learners can query for the joint state-action value function, and one for DT (i.e., $\mathbf{g}_D^i$), where every agent can only query for its own state-action value function. These estimators, respectively, are given by

$$\mathbf{g}_C^i = \sum_{t=0}^\infty \gamma^t \hat{Q} \left( s_t, \mathbf{a}_t^{-i}, a_t^i \right) \nabla_{\theta^i} \log \pi_{\boldsymbol{\theta}}^i \left( a_t^i | s_t \right) \quad \text{and} \quad \mathbf{g}_D^i = \sum_{t=0}^\infty \gamma^t \hat{Q}^i \left( s_t, a_t^i \right) \nabla_{\theta^i} \log \pi_{\boldsymbol{\theta}}^i \left( a_t^i | s_t \right),$$

where $s_t \sim d_{\boldsymbol{\theta}}^t$, $a_t^i \sim \pi_{\boldsymbol{\theta}}^i(\cdot|s_t)$, $\mathbf{a}_t^{-i} \sim \boldsymbol{\pi}_{\boldsymbol{\theta}}^{-i}(\cdot|s_t)$. Here, $\hat{Q}$ is a (joint) critic which agents query for values of $Q_{\boldsymbol{\theta}}(s, \mathbf{a})$. Similarly, $\hat{Q}^i$ is a critic providing values of $Q_{\boldsymbol{\theta}}^i(s, a^i)$. The roles of the critics are, in practice, played by neural networks that can be trained with TD-learning [7, 34]. For the purpose of this paper, we assume they give exact values. In CTDE, a *baseline* [7] is any function $b(s, \mathbf{a}^{-i})$. For any such function, we can easily prove that

$$\mathbb{E}_{s \sim d_{\boldsymbol{\theta}}^t, \mathbf{a}^{-i} \sim \boldsymbol{\pi}_{\boldsymbol{\theta}}^{-i}, a^i \sim \pi_{\boldsymbol{\theta}}^i} \left[ b(s, \mathbf{a}^{-i}) \nabla_{\theta^i} \log \pi_{\boldsymbol{\theta}}^i(a^i|s) \right] = \mathbf{0},$$

(for proof see Appendix A or [7]) which allows augmenting the CTDE estimator as follows

$$\mathbf{g}_C^i(b) = \sum_{t=0}^\infty \gamma^t \left[ \hat{Q}(s_t, \mathbf{a}_t^{-i}, a_t^i) - b(s_t, \mathbf{a}_t^{-i}) \right] \nabla_{\theta^i} \log \pi_{\boldsymbol{\theta}}^i(a_t^i|s_t), \tag{2}$$

---

[(1)]It could have been more clear to write $\pi_{\theta^i}^i$ to highlight that $\pi^i$ depends only on $\theta^i$ and no other parts of $\boldsymbol{\theta}$. Our notation, however, allows for more convenience in the later algebraic manipulations.

[(2)]The notation "$\mathbf{a}$" and "$(\mathbf{a}^{-i}, a^i)$", as well as "$\mathbf{a} \sim \boldsymbol{\pi}_{\boldsymbol{\theta}}$" and "$\mathbf{a}^{-i} \sim \boldsymbol{\pi}_{\boldsymbol{\theta}}^{-i}, a^i \sim \pi_{\boldsymbol{\theta}}^i$", are equivalent. We write $a^i$ and $\boldsymbol{a}$ when we refer to the action and joint action as to values, and $a^i$ and $\mathbf{a}$ as to random variables.

and it has exactly the same expectation as $\mathbf{g}_C^i = \mathbf{g}_C^i(0)$, but can lead to different variance properties. In this paper, we study total variance, which is the sum of variances of all components $\mathbf{g}_{C,j}^i(b)$ of a vector estimator $\mathbf{g}_C^i(b)$. We note that one could consider the variance of every component of parameters, and for each of them choose a tuned baseline [23]. This, however, in light of neural networks with overwhelming parameter sizes used in deep MARL seems not to have practical applications.

## 2.3 Existing CTDE Methods

The first stream of CTDE methods uses the collected experience in order to approximate MAPG and apply stochastic gradient ascent [34] to optimise the policy parameters.

**COMA** [7] is one of the most successful examples of these. It employs a centralised critic, which it adopts to compute a counterfactual baseline $b(s, \mathbf{a}^{-i}) = \hat{Q}^{-i}(s, \mathbf{a}^{-i})$. Together with Equations 1 & 2, COMA gives the following MAPG estimator

$$\mathbf{g}_{\text{COMA}}^i = \sum_{t=0}^{\infty} \gamma^t \hat{A}^i(s_t, \mathbf{a}_t^{-i}, a_t^i) \nabla_{\theta^i} \log \pi_{\boldsymbol{\theta}}^i(a_t^i|s_t) \tag{3}$$

Another stream is the one of trust-region methods, started in RL by TRPO [27] in which at every iteration, the algorithm aims to maximise the total reward with a policy in proximity of the current policy. It achieves it by maximising the objective

$$\mathbb{E}_{s \sim \rho_{\theta_{\text{old}}}, a \sim \pi_{\theta_{\text{old}}}} \left[ \frac{\pi_\theta(a|s)}{\pi_{\theta_{\text{old}}}(a|s)} \hat{A}(s, a) \right], \quad \text{subject to } \mathbb{E}_{s \sim \rho_{\theta_{\text{old}}}} \left[ D_{\text{KL}} \left( \pi_{\theta_{\text{old}}}(\cdot|s) \, \| \, \pi_\theta(\cdot|s) \right) \right] \leq \delta. \tag{4}$$

PPO [29] has been developed as a trust-region method that is friendly for implementation; it approximates TRPO by implementing the constrained optimisation by means of the PPO-clip objective.

**Multi-Agent PPO.** PPO methods can be naturally extended to the MARL setting by leveraging the CTDE framework to train a shared policy for each agent via maximising the sum of their PPO-clip objectives, written as

$$\sum_{i=1}^n \mathbb{E}_{s \sim \rho_{\boldsymbol{\theta}_{\text{old}}}, \mathbf{a} \sim \boldsymbol{\pi}_{\boldsymbol{\theta}_{\text{old}}}} \left[ \min \left( \frac{\pi_{\boldsymbol{\theta}}(a^i|s)}{\pi_{\boldsymbol{\theta}_{\text{old}}}(a^i|s)} \hat{A}(s, \mathbf{a}), \text{ clip} \left( \frac{\pi_{\boldsymbol{\theta}}(a^i|s)}{\pi_{\boldsymbol{\theta}_{\text{old}}}(a^i|s)}, 1 - \epsilon, 1 + \epsilon \right) \hat{A}(s, \mathbf{a}) \right) \right]. \tag{5}$$

The clip operator replaces the ratio $\frac{\pi_{\boldsymbol{\theta}}(a^i|s)}{\pi_{\boldsymbol{\theta}_{\text{old}}}(a^i|s)}$ with, $1 - \epsilon$ when its value is lower, or $1 + \epsilon$ when its value is higher, to prevent large policy updates. Existing implementations of Equation 5 have been mentioned by [4, 45].

In addition to these, there are other streams of CTDE methods in MARL, such as MADDPG [15], which follow the idea of deterministic policy gradient method [30], and QMIX [24], Q-DPP [44] and FQL [49], which focus on the value function decomposition. These methods learn either a deterministic policy or a value function, thus are not in the scope of stochastic MAPG methods.

## 3 Analysis and Improvement of Multi-agent Policy Gradient Estimates

In this section, we provide a detailed analysis of the variance of the MAPG estimator, and propose a method for its reduction. Throughout the whole section we rely on the following two assumptions.

**Assumption 1.** *The state space $\mathcal{S}$, and every agent $i$'s action space $\mathcal{A}^i$ is either discrete and finite, or continuous and compact.*

**Assumption 2.** *For all $i \in \mathcal{N}$, $s \in \mathcal{S}$, $a^i \in \mathcal{A}^i$, the map $\theta^i \mapsto \pi_{\boldsymbol{\theta}}^i(a^i|s)$ is continuously differentiable.*

### 3.1 Analysis of MAPG Variance

The goal of this subsection is to demonstrate how an agent's MAPG estimator's variance is influenced by other agents. In particular, we show how the presence of other agents makes the MARL problem different from single-agent RL problems (e.g., the DT framework). We start our analysis by studying the variance of the component in CTDE estimator that depends on other agents, which is the joint $Q$-function. Since $\mathbf{Var}_{\mathbf{a} \sim \boldsymbol{\pi}_{\boldsymbol{\theta}}} [Q_{\boldsymbol{\theta}}(s, \mathbf{a})] = \mathbf{Var}_{\mathbf{a} \sim \boldsymbol{\pi}_{\boldsymbol{\theta}}} [V_{\boldsymbol{\theta}}(s) + A_{\boldsymbol{\theta}}(s, \mathbf{a})] = \mathbf{Var}_{\mathbf{a} \sim \boldsymbol{\pi}_{\boldsymbol{\theta}}} [A_{\boldsymbol{\theta}}(s, \mathbf{a})]$, we

can focus our analysis on the advantage function, which has more interesting algebraic properties. We start by presenting a simple lemma which, in addition to being a premise for the main result, offers some insights about the relationship between RL and MARL problems.

**Lemma 1** (Multi-agent advantage decomposition). *For any state $s \in \mathcal{S}$, the following equation holds for any subset of $m$ agents and any permutation of their labels,*

$$A_{\boldsymbol{\theta}}^{1,\ldots,m}\left(s, \boldsymbol{a}^{(1,\ldots,m)}\right) = \sum_{i=1}^{m} A_{\boldsymbol{\theta}}^{i}\left(s, \boldsymbol{a}^{(1,\ldots,i-1)}, a^{i}\right).$$

For proof see Appendix B.1. The statement of this lemma is that the joint advantage of agents' joint action is the sum of sequentially unfolding multi-agent advantages of individual agents' actions. It suggests that a MARL problem can be considered as a sum of $n$ RL problems. The intuition from Lemma 1 leads to an idea of decomposing the variance of the total advantage into variances of multi-agent advantages of individual agents. Leveraging the proof of Lemma 1, we further prove that

**Lemma 2.** *For any state $s \in \mathcal{S}$, we have*

$$\mathbf{Var}_{\mathbf{a} \sim \boldsymbol{\pi_\theta}}\left[A_{\boldsymbol{\theta}}(s, \mathbf{a})\right] = \sum_{i=1}^{n} \mathbb{E}_{\mathbf{a}^1 \sim \pi_{\boldsymbol{\theta}}^1, \ldots, \mathbf{a}^{i-1} \sim \pi_{\boldsymbol{\theta}}^{i-1}}\left[\mathbf{Var}_{\mathbf{a}^i \sim \pi_{\boldsymbol{\theta}}^i}\left[A_{\boldsymbol{\theta}}^i\left(s, \mathbf{a}^{(1,\ldots,i-1)}, \mathbf{a}^i\right)\right]\right].$$

For proof see Appendix B.1. The above result reveals that the variance of the total advantage takes a sequential and additive structure of the advantage that is presented in Lemma 1. This hints that a similar additive relation can hold once we loose the sequential structure of the multi-agent advantages. Indeed, the next lemma, which follows naturally from Lemma 2, provides an upper bound for the joint advantage's variance in terms of local advantages, and establishes a notion of additivity of variance in MARL.

**Lemma 3.** *For any state $s \in \mathcal{S}$, we have*

$$\mathbf{Var}_{\mathbf{a} \sim \boldsymbol{\pi_\theta}}\left[A_{\boldsymbol{\theta}}(s, \mathbf{a})\right] \leq \sum_{i=1}^{n} \mathbf{Var}_{\mathbf{a}^{-i} \sim \boldsymbol{\pi_\theta}^{-i}, \mathbf{a}^i \sim \pi_{\boldsymbol{\theta}}^i}\left[A_{\boldsymbol{\theta}}^i(s, \mathbf{a}^{-i}, \mathbf{a}^i)\right].$$

For proof see Appendix B.1. Upon these lemmas we derive the main theoretical result of this subsection. The following theorem describes the order of excess variance that the centralised policy gradient estimator has over the decentralised one.

**Theorem 1.** *The CTDE and DT estimators of MAPG satisfy*

$$\mathbf{Var}_{s_{0:\infty} \sim d_{\boldsymbol{\theta}}^{0:\infty}, \mathbf{a}_{0:\infty} \sim \boldsymbol{\pi_\theta}}\left[\mathbf{g}_C^i\right] - \mathbf{Var}_{s_{0:\infty} \sim d_{\boldsymbol{\theta}}^{0:\infty}, \mathbf{a}_{0:\infty} \sim \boldsymbol{\pi_\theta}}\left[\mathbf{g}_D^i\right] \leq \frac{B_i^2}{1 - \gamma^2} \sum_{j \neq i} \epsilon_j^2 \leq (n-1)\frac{(\epsilon B_i)^2}{1 - \gamma^2}$$

*where $B_i = \sup_{s,\mathbf{a}} \left|\left|\nabla_{\theta^i} \log \pi_{\boldsymbol{\theta}}^i\left(\mathbf{a}^i | s\right)\right|\right|$, $\epsilon_i = \sup_{s, \boldsymbol{a}^{-i}, a^i} \left|A_{\boldsymbol{\theta}}^i(s, \boldsymbol{a}^{-i}, a^i)\right|$, and $\epsilon = \max_i \epsilon_i$.*

*Proof sketch.* (For the full proof see Appendix B.2.) We start the proof by fixing a state $s$ and considering the difference of $\mathbf{Var}_{\mathbf{a} \sim \boldsymbol{\pi_\theta}}[\hat{Q}(s, \mathbf{a}^{-i}, \mathbf{a}^i)\nabla_{\theta^i} \log \pi_{\boldsymbol{\theta}}^i(\mathbf{a}^i|s)] - \mathbf{Var}_{\mathbf{a}^i \sim \pi_{\boldsymbol{\theta}}^i}[\hat{Q}^i(s, \mathbf{a}^i)\nabla_{\theta^i} \log \pi_{\boldsymbol{\theta}}^i(\mathbf{a}^i|s)]$. The goal of our proof strategy was to collate the terms $\hat{Q}(s, \mathbf{a}^{-i}, \mathbf{a}^i)$ and $\hat{Q}^i(s, \mathbf{a}^i)$ because such an expression could be related to the above lemmas about multi-agent advantage, given that the latter quantity is the expected value of the former when $\mathbf{a}^{-i} \sim \pi_{\boldsymbol{\theta}}^{-i}$. Based on the fact that these two estimators are unbiased, we transform the considered difference into $\mathbb{E}_{\mathbf{a} \sim \boldsymbol{\pi_\theta}}\left[\left|\left|\nabla_{\theta^i} \log \pi_{\boldsymbol{\theta}}^i(\mathbf{a}^i|s)\right|\right|^2 \left(\hat{Q}(s, \mathbf{a}) - \hat{Q}^i(s, \mathbf{a}^i)\right)^2\right]$. Using the upper bound $B_i$, the fact that $\hat{A}^{-i}(s, \mathbf{a}^i, \mathbf{a}^{-i}) = \hat{Q}(s, \mathbf{a}^i, \mathbf{a}^{-i}) - \hat{Q}^i(s, \mathbf{a}^i)$, and the result of Lemma 3, we bound this expectation by $B_i^2 \sum_{j \neq i} \mathbb{E}_{\mathbf{a}^i \sim \pi_{\boldsymbol{\theta}}^i}\left[\mathbf{Var}_{\mathbf{a}^{-i} \sim \pi_{\boldsymbol{\theta}}^{-i}}\left[\hat{A}^j(s, \mathbf{a}^{-j}, \mathbf{a}^j)\right]\right]$, which we then rewrite to bound it by $B_i^2 \sum_{j \neq i} \epsilon_j^2$. As such, the first inequality in the theorem follows from summing, with discounting, over all time steps $t$, and the second one is its trivial upper bound. $\qquad\square$

The result in Theorem 1 exposes the level of difference between MAPG and PG estimation which, measured by variance, is not only non-negative, as shown in [16], but can grow **linearly** with the number of agents. More precisely, a CTDE learner's gradient estimator comes with an extra price of variance coming from other agents' local advantages (i.e., explorations). This further suggests

that we shall search for variance reduction techniques which augment the state-action value signal. In RL, such a well-studied technique is baseline-subtraction, where many successful baselines are state-dependent. In CTDE, we can employ baselines taking state and other agents' actions into account. We demonstrate their strength on the example of a counterfactual baseline of COMA [7].

**Theorem 2.** *The COMA and DT estimators of MAPG satisfy*

$$\mathbf{Var}_{s_{0:\infty} \sim d_{\boldsymbol{\theta}}^{0:\infty}, \mathbf{a}_{0:\infty} \sim \boldsymbol{\pi}_{\boldsymbol{\theta}}} \left[ \mathbf{g}_{COMA}^i \right] - \mathbf{Var}_{s_{0:\infty} \sim d_{\boldsymbol{\theta}}^{0:\infty}, \mathbf{a}_{0:\infty} \sim \boldsymbol{\pi}_{\boldsymbol{\theta}}} \left[ \mathbf{g}_D^i \right] \; \leq \; \frac{(\epsilon_i B_i)^2}{1 - \gamma^2}$$

For proof see Appendix B.2. The above theorem discloses the effectiveness of the counterfactual baseline. COMA baseline essentially allows to drop the number of agents from the order of excess variance of CTDE, thus potentially binding it closely to the single-agent one. Yet, such binding is not exact, since it still contains the dependence on the local advantage, which can be very large in scenarios when, for example, a single agent has a chance to revert its collaborators' errors with its own single action. Based on such insights, in the following subsections, we study the method of optimal baselines, and derive a solution to these issues above.

## 3.2 The Optimal Baseline for MAPG

In order to search for the optimal baseline, we first demonstrate how it can impact the variance of an MAPG estimator. To achieve that, we decompose the variance at an arbitrary time step $t \geq 0$, and separate the terms which are subject to possible reduction from those unchangable ones. Let us denote $\mathbf{g}_{C,t}^i(b) = [\hat{Q}(s_t, \mathbf{a}_t) - b]\nabla_{\theta^i} \log \pi_{\boldsymbol{\theta}}^i(\mathbf{a}_t^i | s_t)$, sampled with $s_t \sim d_{\boldsymbol{\theta}}^t, \mathbf{a}_t \sim \boldsymbol{\pi}_{\boldsymbol{\theta}}(\cdot | s_t)$, which is essentially the $t^{\text{th}}$ summand of the gradient estimator given by Equation 2. Specifically, we have

$$\mathbf{Var}_{s_t \sim d_{\boldsymbol{\theta}}^t, \mathbf{a}_t \sim \boldsymbol{\pi}_{\boldsymbol{\theta}}} \left[ \mathbf{g}_{C,t}^i(b) \right] = \mathbf{Var}_{s_t \sim d_{\boldsymbol{\theta}}^t} \left[ \mathbb{E}_{\mathbf{a}_t \sim \boldsymbol{\pi}_{\boldsymbol{\theta}}} \left[ \mathbf{g}_{C,t}^i(b) \right] \right] + \mathbb{E}_{s_t \sim d_{\boldsymbol{\theta}}^t} \left[ \mathbf{Var}_{\mathbf{a}_t \sim \boldsymbol{\pi}_{\boldsymbol{\theta}}} \left[ \mathbf{g}_{C,t}^i(b) \right] \right] \quad (6)$$

$$= \mathbf{Var}_{s_t \sim d_{\boldsymbol{\theta}}^t} \left[ \mathbb{E}_{\mathbf{a}_t \sim \boldsymbol{\pi}_{\boldsymbol{\theta}}} \left[ \mathbf{g}_{C,t}^i(b) \right] \right] + \mathbb{E}_{s_t \sim d_{\boldsymbol{\theta}}^t} \left[ \mathbf{Var}_{\mathbf{a}_t^{-i} \sim \boldsymbol{\pi}_{\boldsymbol{\theta}}^{-i}} \left[ \mathbb{E}_{\mathbf{a}_t^i \sim \boldsymbol{\pi}_{\boldsymbol{\theta}}^i} \left[ \mathbf{g}_{C,t}^i(b) \right] \right] + \mathbb{E}_{\mathbf{a}_t^{-i} \sim \boldsymbol{\pi}_{\boldsymbol{\theta}}^{-i}} \left[ \mathbf{Var}_{\mathbf{a}_t^i \sim \boldsymbol{\pi}_{\boldsymbol{\theta}}^i} \left[ \mathbf{g}_{C,t}^i(b) \right] \right] \right]$$

$$= \underbrace{\mathbf{Var}_{s_t \sim d_{\boldsymbol{\theta}}^t} \left[ \mathbb{E}_{\mathbf{a}_t \sim \boldsymbol{\pi}_{\boldsymbol{\theta}}} \left[ \mathbf{g}_{C,t}^i(b) \right] \right]}_{\text{Variance from state}} + \underbrace{\mathbb{E}_{s_t \sim d_{\boldsymbol{\theta}}^t} \left[ \mathbf{Var}_{\mathbf{a}_t^{-i} \sim \boldsymbol{\pi}_{\boldsymbol{\theta}}^{-i}} \left[ \mathbb{E}_{\mathbf{a}_t^i \sim \boldsymbol{\pi}_{\boldsymbol{\theta}}^i} \left[ \mathbf{g}_{C,t}^i(b) \right] \right] \right]}_{\text{Variance from other agents' actions}} + \underbrace{\mathbb{E}_{s_t \sim d_{\boldsymbol{\theta}}^t, \mathbf{a}_t^{-i} \sim \boldsymbol{\pi}_{\boldsymbol{\theta}}^{-i}} \left[ \mathbf{Var}_{\mathbf{a}_t^i \sim \boldsymbol{\pi}_{\boldsymbol{\theta}}^i} \left[ \mathbf{g}_{C,t}^i(b) \right] \right]}_{\text{Variance from agent } i\text{'s action}}.$$

Thus, in a CTDE estimator, there are three main sources of variance, which are: state $s_t$, other agents' joint action $\mathbf{a}_t^{-i}$, and the agent $i$'s action $\mathbf{a}_t^i$. The first two terms of the right-hand side of the above equation, which are those involving variance coming from $s_t$ and $\mathbf{a}_t^{-i}$, remain constant for all $b$. However, the baseline subtraction influences the local variance of the agent, $\mathbf{Var}_{\mathbf{a}_t^i \sim \pi_{\boldsymbol{\theta}}^i} \left[ \mathbf{g}_{C,t}^i(b) \right]$, and therefore minimising it for every $(s, \mathbf{a}^{-i})$ pair minimises the third term, which is equivalent to minimising the entire variance. In this subsection, we describe how to perform this minimisation.

**Theorem 3** (Optimal baseline for MAPG). *The optimal baseline (OB) for the MAPG estimator is*

$$b^{optimal}(s, \mathbf{a}^{-i}) = \frac{\mathbb{E}_{\mathbf{a}^i \sim \pi_{\boldsymbol{\theta}}^i} \left[ \hat{Q}(s, \mathbf{a}^{-i}, \mathbf{a}^i) \left|\left| \nabla_{\theta^i} \log \pi_{\boldsymbol{\theta}}^i(\mathbf{a}^i | s) \right|\right|^2 \right]}{\mathbb{E}_{\mathbf{a}^i \sim \pi_{\boldsymbol{\theta}}^i} \left[ \left|\left| \nabla_{\theta^i} \log \pi_{\boldsymbol{\theta}}^i(\mathbf{a}^i | s) \right|\right|^2 \right]} \quad (7)$$

For proof see Appendix C.1. Albeit elegant, OB in Equation 7 is computationally challenging to estimate due to the fact that it requires a repeated computation of the norm of the gradient $\nabla_{\theta^i} \log \pi_{\boldsymbol{\theta}}^i(\mathbf{a}^i | s)$, which can have dimension of order $\sim 10^4$ when the policy is parametrised by a neural network (e.g., see [31, Appendix C.2]). Furthermore, in continuous action spaces, as in principle the $Q$-function does not have a simple analytical form, this baseline cannot be computed exactly, and instead it must be approximated. This is problematic, too, because the huge dimension of the gradients may induce large variance in the approximation of OB in addition to the variance of the policy gradient estimation. To make OB computable and applicable, we formulate in the next section a surrogate variance-minimisation objective, whose solution is much more tractable.

## 3.3 Optimal Baselines for Deep Neural Networks

Recall that in deep MARL, the policy $\pi_{\boldsymbol{\theta}}^i$ is assumed to be a member of a specific family of distributions, and the network $\theta^i$ only computes its parameters, which we can refer to as $\psi_{\boldsymbol{\theta}}^i$. In

the discrete MARL, it can be the last layer before $\mathrm{softmax}$, and in the continuous MARL, $\psi_{\boldsymbol{\theta}}^i$ can be the mean and the standard deviation of a Gaussian distribution [7, 45]. We can then write $\pi_{\boldsymbol{\theta}}^i(\mathrm{a}^i|\mathrm{s}) = \pi^i\left(\mathrm{a}^i|\psi_{\boldsymbol{\theta}}^i(\mathrm{s})\right)$, and factorise the gradient $\nabla_{\theta^i}\log\pi_{\boldsymbol{\theta}}^i$ with the chain rule. This allows us to rewrite the local variance as

$$
\begin{aligned}
&\mathbf{Var}_{\mathrm{a}^i\sim\pi_{\boldsymbol{\theta}}^i}\left[\nabla_{\theta^i}\log\pi_{\boldsymbol{\theta}}^i\left(\mathrm{a}^i|\psi_{\boldsymbol{\theta}}^i(\mathrm{s})\right)\left(\hat{Q}(\mathrm{s},\mathbf{a}^{-i},\mathrm{a}^i)-b(\mathrm{s},\mathbf{a}^{-i})\right)\right]\\
=&\mathbf{Var}_{\mathrm{a}^i\sim\pi_{\boldsymbol{\theta}}^i}\left[\nabla_{\theta^i}\psi_{\boldsymbol{\theta}}^i(\mathrm{s})\nabla_{\psi_{\boldsymbol{\theta}}^i(\mathrm{s})}\log\pi^i\left(\mathrm{a}^i|\psi_{\boldsymbol{\theta}}^i(\mathrm{s})\right)\left(\hat{Q}(\mathrm{s},\mathbf{a}^{-i},\mathrm{a}^i)-b(\mathrm{s},\mathbf{a}^{-i})\right)\right]\\
=&\nabla_{\theta^i}\psi_{\boldsymbol{\theta}}^i(\mathrm{s})\mathbf{Var}_{\mathrm{a}^i\sim\pi_{\boldsymbol{\theta}}^i}\left[\nabla_{\psi_{\boldsymbol{\theta}}^i(\mathrm{s})}\log\pi^i\left(\mathrm{a}^i|\psi_{\boldsymbol{\theta}}^i(\mathrm{s})\right)\left(\hat{Q}(\mathrm{s},\mathbf{a}^{-i},\mathrm{a}^i)-b(\mathrm{s},\mathbf{a}^{-i})\right)\right]\nabla_{\theta^i}\psi_{\boldsymbol{\theta}}^i(\mathrm{s})^T. \quad (8)
\end{aligned}
$$

This allows us to formulate a surrogate minimisation objective, which is the variance term from Equation 8, which we refer to as *surrogate local variance*. The optimal baseline for this objective comes as a corollary to the proof of Theorem 3.

**Corollary 1.** *The optimal baseline for the surrogate local variance in Equation 8 is*

$$
b^*(\mathrm{s},\mathbf{a}^{-i}) = \frac{\mathbb{E}_{\mathrm{a}^i\sim\pi_{\boldsymbol{\theta}}^i}\left[\hat{Q}(\mathrm{s},\mathbf{a}^{-i},\mathrm{a}^i)\left|\left|\nabla_{\psi_{\boldsymbol{\theta}}^i(\mathrm{s})}\log\pi^i\left(\mathrm{a}^i|\psi_{\boldsymbol{\theta}}^i(\mathrm{s})\right)\right|\right|^2\right]}{\mathbb{E}_{\mathrm{a}^i\sim\pi_{\boldsymbol{\theta}}^i}\left[\left|\left|\nabla_{\psi_{\boldsymbol{\theta}}^i(\mathrm{s})}\log\pi^i\left(\mathrm{a}|\psi_{\boldsymbol{\theta}}^i(\mathrm{s})\right)\right|\right|^2\right]}. \quad (9)
$$

Note that the vector $\nabla_{\psi_{\boldsymbol{\theta}}^i(\mathrm{s})}\log\pi^i\left(\mathrm{a}^i|\psi_{\boldsymbol{\theta}}^i(\mathrm{s})\right)$ can be computed without backpropagation when the family of distributions to which $\pi_{\boldsymbol{\theta}}^i$ belongs is known, which is fairly common in deep MARL. Additionally, the dimension of this vector is of the same size as the size of the action space, which is in the order $\sim 10$ in many cases (e.g., [7, 32]) which makes computations tractable. Equation 9 essentially allows us to incorporate the OB in any existing (deep) MAPG methods, accouting for both continuous and discrete-action taks. Hereafater, we refer to the surrogate OB in Equation 9 as the OB and apply it in the later experiment section.

### 3.4 Excess Variance of MAPG/COMA *vs.* OB

We notice that for a probability measure $x_{\psi^i}^i(\mathrm{a}^i|s) = \frac{\pi_{\boldsymbol{\theta}}^i(\mathrm{a}^i|s)\left|\left|\nabla_{\psi_{\boldsymbol{\theta}}^i(s)}\log\pi^i(\mathrm{a}^i|\psi_{\boldsymbol{\theta}}^i(s))\right|\right|^2}{\mathbb{E}_{\mathrm{a}^i\sim\pi_{\boldsymbol{\theta}}^i}\left[\left|\left|\nabla_{\psi_{\boldsymbol{\theta}}^i(s)}\log\pi^i(\mathrm{a}^i|\psi_{\boldsymbol{\theta}}^i(s))\right|\right|^2\right]}$, the OB takes the form of $b^*(\mathrm{s},\mathbf{a}^{-i}) = \mathbb{E}_{\mathrm{a}^i\sim x_{\psi_{\boldsymbol{\theta}}^i}^i}\left[\hat{Q}(\mathrm{s},\mathbf{a}^{-i},\mathrm{a}^i)\right]$. It is then instructive to look at a practical deep MARL example, which is that of a discrete actor with policy $\pi_{\boldsymbol{\theta}}^i(\mathrm{a}^i|\mathrm{s}) = \mathrm{softmax}(\psi_{\boldsymbol{\theta}}^i(\mathrm{s}))(\mathrm{a}^i)$. In this case, we can derive that

$$
x_{\psi_{\boldsymbol{\theta}}^i}^i\left(\mathrm{a}^i|\mathrm{s}\right) \propto \pi_{\boldsymbol{\theta}}^i\left(\mathrm{a}^i|\mathrm{s}\right)\left(1+\left|\left|\pi_{\boldsymbol{\theta}}^i(\mathrm{s})\right|\right|^2 - 2\pi_{\boldsymbol{\theta}}^i\left(\mathrm{a}^i|\mathrm{s}\right)\right) \quad (10)
$$

(Full derivation is shown in Appendix C.2). This measure, in contrast to COMA, scales up the weight of actions with small weight in $\pi_{\boldsymbol{\theta}}^i$, and scales down the weight of actions with large $\pi_{\boldsymbol{\theta}}^i$, while COMA's baseline simply takes each action with weight determined by $\pi_{\boldsymbol{\theta}}^i$, which has an opposite effect to OB. Let *the excess surrogate local variance* of a CTDE MAPG estimator $\mathbf{g}_{\mathrm{C}}^i(b)$ of agent $i$ be defined as Equation 11. We analyse the excess variance in Theorem 4.

$$
\Delta\mathbf{Var}(b) \triangleq \mathbf{Var}_{\mathrm{a}^i\sim\pi_{\boldsymbol{\theta}}^i}\left[\mathbf{g}_{\mathrm{C}}^i(b)\right] - \mathbf{Var}_{\mathrm{a}^i\sim\pi_{\boldsymbol{\theta}}^i}\left[\mathbf{g}_{\mathrm{C}}^i(b^*)\right] \quad (11)
$$

**Theorem 4.** *The excess surrogate local variance for baseline $b$ satisfies*

$$
\Delta\mathbf{Var}(b) = \left(b - b^*(\mathrm{s},\boldsymbol{a}^{-i})\right)^2\mathbb{E}_{\mathrm{a}^i\sim\pi_{\boldsymbol{\theta}}^i}\left[\left|\left|\nabla_{\psi_{\boldsymbol{\theta}}^i}\log\pi^i\left(\mathrm{a}^i|\psi_{\boldsymbol{\theta}}^i(\mathrm{s})\right)\right|\right|^2\right]
$$

*In particular, the excess variance of the vanilla MAPG and COMA estimators satisfy*

$$
\Delta\mathbf{Var}_{MAPG} \leq D_i^2\left(\mathbf{Var}_{\mathrm{a}^i\sim\pi_{\boldsymbol{\theta}}^i}[A_{\boldsymbol{\theta}}^i(s,\boldsymbol{a}^{-i},\mathrm{a}^i)] + Q_{\boldsymbol{\theta}}^{-i}(s,\boldsymbol{a}^{-i})^2\right) \leq D_i^2\left(\epsilon_i^2 + \left[\frac{\beta}{1-\gamma}\right]^2\right)
$$

$$
\Delta\mathbf{Var}_{COMA} \leq D_i^2\,\mathbf{Var}_{\mathrm{a}^i\sim\pi_{\boldsymbol{\theta}}^i}\left[A_{\boldsymbol{\theta}}^i(s,\boldsymbol{a}^{-i},\mathrm{a}^i)\right] \leq (\epsilon_i D_i)^2
$$

*where $D_i = \sup_{\mathrm{a}^i}\left|\left|\nabla_{\psi_{\boldsymbol{\theta}}^i}\log\pi_{\boldsymbol{\theta}}^i\left(\mathrm{a}^i|\psi_{\boldsymbol{\theta}}^i(\mathrm{s})\right)\right|\right|$, and $\epsilon_i = \sup_{s,\boldsymbol{a}^{-i},\mathrm{a}^i}\left|A_{\boldsymbol{\theta}}^i(s,\boldsymbol{a}^{-i},\mathrm{a}^i)\right|$.*

Table 1: A numerial toy exmaple that shows the effectiveness of OB. For all actions in column $\mathrm{a}^i$, agent $i$ is provided with the last layer before $\mathrm{softmax}$ of its actor network and the actions' values (columns $\psi^i_{\boldsymbol{\theta}}(\mathrm{a}^i)$ and $\hat{Q}(\boldsymbol{a}^{-i}, \mathrm{a}^i)$). It computes the remaining quantities in the table, which are used to derive the three gradient estimators, whose variance is summarised in the right part of the table. The full calculations of the below values are stored in Appendix E.

| $\mathrm{a}^i$ | $\psi^i_{\boldsymbol{\theta}}(\mathrm{a}^i)$ | $\pi^i_{\boldsymbol{\theta}}(\mathrm{a}^i)$ | $x^i_{\psi^i_{\boldsymbol{\theta}}}(\mathrm{a}^i)$ | $\hat{Q}(\boldsymbol{a}^{-i}, \mathrm{a}^i)$ | $\hat{A}^i(\boldsymbol{a}^{-i}, \mathrm{a}^i)$ | $\hat{X}^i(\boldsymbol{a}^{-i}, \mathrm{a}^i)$ | Method | Variance |
|---|---|---|---|---|---|---|---|---|
| 1 | $\log 8$ | 0.8 | 0.14 | 2 | $-9.7$ | $-41.71$ | MAPG | **1321** |
| 2 | 0 | 0.1 | 0.43 | 1 | $-10.7$ | $-42.71$ | COMA | **1015** |
| 3 | 0 | 0.1 | 0.43 | 100 | 88.3 | 56.29 | OB | **673** |

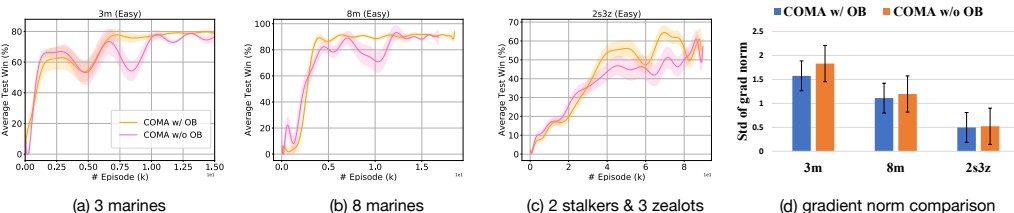

(a) 3 marines     (b) 8 marines     (c) 2 stalkers & 3 zealots     (d) gradient norm comparison

Figure 1: Performance comparisons between COMA with and without OB on three SMAC challenges.

For proof see Appendix C.3. This theorem implies that OB is particularly helpful in situations when the value of $Q^{-i}$ function is large, or when agent $i$'s local advantage has large variance. In these scenarios, a baseline like COMA might fail because when certain actions have large local advantage, we would want the agent to learn them, although its gradient estimate may be inaccurate, disabling the agent to learn to take the action efficiently.

### 3.5 Implementation of the Optimal Baseline

Our OB technique is a general method to any MAPG methods with a joint critic. It can be seamlessly integrated into any existing MARL algorithms that require MAPG estimators. One only needs to replace the algorithm's state-action value signal (either state-action value or advantage function) with

$$\hat{X}^i(\mathrm{s}, \mathbf{a}^{-i}, \mathrm{a}^i) = \hat{Q}(\mathrm{s}, \mathbf{a}^{-i}, \mathrm{a}^i) - b^*(\mathrm{s}, \mathbf{a}^{-i}). \tag{12}$$

This gives us an estimator of **COMA with OB**

$$\mathbf{g}^i_\mathrm{X} = \sum_{t=0}^{\infty} \gamma^t \hat{X}^i(\mathrm{s}_t, \mathbf{a}_t^{-i}, \mathrm{a}_t^i)\nabla_{\theta^i} \log \pi^i_{\boldsymbol{\theta}}(\mathrm{a}_t^i|\mathrm{s}_t),$$

and a variant of **Multi-agent PPO with OB**, which maximises the objective of

$$\sum_{i=1}^n \mathbb{E}_{\mathrm{s} \sim \rho_{\boldsymbol{\theta}_{old}}, \mathbf{a} \sim \boldsymbol{\pi}_{\boldsymbol{\theta}_{old}}} \left[ \min \left( \frac{\pi_{\boldsymbol{\theta}}(\mathrm{a}^i|\mathrm{s})}{\pi_{\boldsymbol{\theta}_{old}}(\mathrm{a}^i|\mathrm{s})} \hat{X}^i(\mathrm{s}, \mathbf{a}), \ \mathrm{clip}\left( \frac{\pi_{\boldsymbol{\theta}}(\mathrm{a}^i|\mathrm{s})}{\pi_{\boldsymbol{\theta}_{old}}(\mathrm{a}^i|\mathrm{s})}, 1-\epsilon, 1+\epsilon \right) \hat{X}^i(\mathrm{s}, \mathbf{a}) \right) \right].$$

In order to compute OB, agent $i$ follows these two steps: firstly, it evaluates the probability measure $x^i_{\psi^i_{\boldsymbol{\theta}}}$, and then computes the expectation of $\hat{Q}(s, \boldsymbol{a}^{-i}, \mathrm{a}^i)$ over it with a dot product. Such a protocol allows for exact computation of OB when the action space is discrete. When it comes to continuous action space, the first step of evaluating $x^i_{\psi^i_{\boldsymbol{\theta}}}$ relies on sampling actions from agent's policy, which gives us the approximation of OB. To make it clear, we provide PyTorch implmentations of OB in both discrete and continuous settings in Appendix D.

## 4 Experiments

On top of theoretical proofs, in this section, we demonstrate empirical evidence that OB can decrease the variance of MAPG estimators, stabilise training, and most importantly, lead to better performance. To verify the adaptability of OB, we apply OB on both COMA and multi-agent PPO methods as described in Section 3.5. We benchmark the OB-modified algorithms against existing state-of-the-art

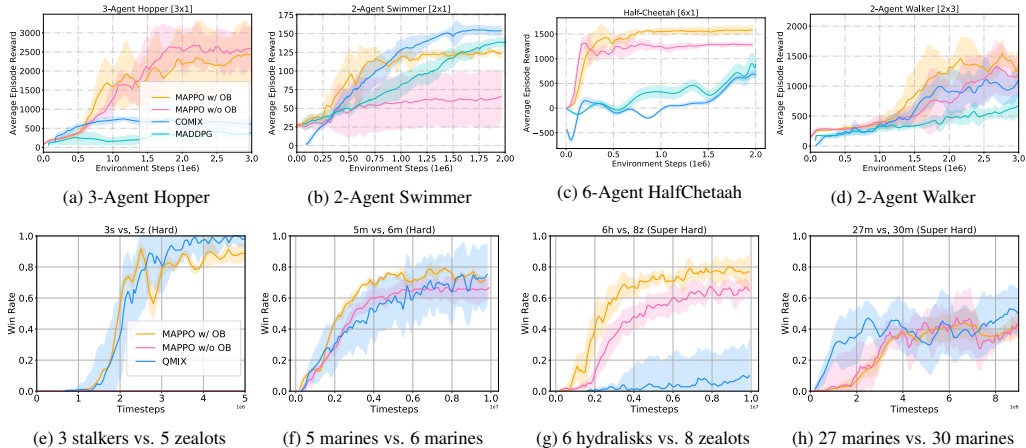

(a) 3-Agent Hopper    (b) 2-Agent Swimmer    (c) 6-Agent HalfChetaah    (d) 2-Agent Walker

(e) 3 stalkers vs. 5 zealots    (f) 5 marines vs. 6 marines    (g) 6 hydralisks vs. 8 zealots    (h) 27 marines vs. 30 marines

Figure 2: Performance comparisons between multi-agent PPO method with and without OB on four multi-agent MuJoCo tasks and four (super-)hard SMAC challenges.

Table 2: Comparisons on the standard deviation of the gradient norm of multi-agent PPO method with and without OB. All quantities are provided in scale $0.01$. Standard errors are provided in brackets. Results suggest OB consistently reduces gradient norms across all eight tasks.

| Method / Task | 3s vs. 5z | 5m vs. 6m | 6h vs. 8z | 27m vs. 30m | 6-Agent HalfCheetah | 3-Agent Hopper | 2-Agent Swimmer | 2-Agent Walker |
|---|---|---|---|---|---|---|---|---|
| MAPPO w/ OB | **2.20 (0.17)** | **1.50 (0.02)** | **2.33 (0.03)** | **11.55 (4.80)** | **30.64 (0.50)** | **79.67 (3.79)** | **66.72 (3.67)** | **372.03 (10.83)** |
| MAPPO w/o OB | 6.67 (0.35) | 2.17 (0.07) | 2.54 (0.09) | 19.62 (6.90) | 33.65 (2.04) | 82.45 (2.79) | 73.54 (11.98) | 405.66 (18.34) |

(SOTA) methods, which include COMA [7] and MAPPO [45], and value-based methods such as QMIX [24] and COMIX [21], and a deterministic PG method, ie., MADDPG [15]. Notably, since OB relies on the Q-function critics, we did not apply the GAE [28] estimator that builds only on the state value function when implementing multi-agent PPO for fair comparisons. For each of the baseline on each task, we report the results of five random seeds. We refer to Appendix F for the detailed hyper-parameter settings for baselines.

**Numerical Toy Example**. We first offer a numerical toy example to demonstrate how the subtraction of OB alters the state-action value signal in the estimator, as well as how this technique performs, against vanilla MAPG and COMA. We assume a stateless setting, and a given joint action of other agents. The results in Table 1 show that the measure $x^i_{\psi^i_\theta}$ puts more weight on actions neglected by $\pi^i_\theta$, lifting the value of OB beyond the COMA baseline, as suggested by Equation 10. The resulting $X^i$ function penalises the sub-optimal actions more heavily. Most importantly, OB provides a MAPG estimator with far lower variance as expected.

**StarCraft Multi-Agent Challenge (SMAC)** [25]. In SMAC, each individual unit is controlled by a learning agent, which has finitely many possible actions to take. The units cooperate to defeat enemy bots across scenarios of different levels of difficulty. Based on Figure 1(a-d), we can tell that OB provides more accurate MAPG estimates and stabilises training of COMA across all three maps. Importantly, COMA with OB learns policies that achieve higher rewards than the classical COMA. Since COMA perform badly on hard and super-hard maps in SMAC [19], we only report their results on easy maps. On the hard and super-hard maps in Figures 2e, 2f, and 2g, OB improves the performance of multi-agent PPO. Surprisingly on Figure 2e, OB improves the winning rate of multi-agent PPO from zero to 90%. Moreover, with an increasing number of agents, the effectiveness of OB increases in terms of offering a low-variance MAPG estimator. According to Table 2, when the tasks involves 27 learning agents, OB offers a 40% reduction in the variance of the gradient norm.

**Multi-Agent MuJoCo** [5]. SMAC are discrete control tasks; here we study the performance of OB when the action space is continuous. In each environment of Multi-Agent MuJoCo, each individual agent controls a part of a shared robot (e.g., a leg of a Hopper), and all agents maximise a shared reward function. Results are consistent with the findings on SMAC in Table 2; on all tasks, OB helps decrease the variance of gradient norm. Moreover, OB can improve the performance of multi-agent PPO on most MuJoCo tasks in Figure 2 (top row), and decreases its variance in all of them. In particular, multi-agent PPO with OB performs the best on Walker (i.e., Figure 2d) and HalfCheetah (i.e., Figure 2c) robots, and with the increasing number of agents, the effectiveness OB becomes apparent; 6-agent HalfCheetah achieves the largest performance gap.

# 5 Conclusion

In this paper, we try to settle the variance of multi-agent policy gradient (MAPG) estimators. We start our contribution by quantifying, for the first time, the variance of the MAPG estimator and revealing the key influencial factors. Specifically, we prove that the excess variance that a centralised estimator has over its decentralised counterpart grows linearly with the number of agents, and quadratically with agents' local advantages. A natural outcome of our analysis is the optimal baseline (OB) technique. We adapt OB to exsiting deep MAPG methods and demonstrate its empirical effectiveness on challenging benchmarks against strong baselines. In the future, we plan to study other variance reduction techniques that can apply without requiring Q-function critics.

## Author Contributions

We summarise the main contributions from each of the authors as follows:

**Jakub Grudzien Kuba**: Theoretical results, algorithm design, code implementation (discrete OB, continuous OB) and paper writing.
**Muning Wen**: Algorithm design, code implementation (discrete OB, continuous OB) and experiments running.
**Linghui Meng**: Code implementation (discrete OB) and experiments running.
**Shangding Gu**: Code implementation (continuous OB) and experiments running.
**Haifeng Zhang**: Computational resources and project discussion.
**David Mguni**: Project discussion.
**Jun Wang**: Project discussion and overall technical supervision.
**Yaodong Yang**: Project lead, idea proposing, theory development and experiment design supervision, and whole manuscript writing. Work were done at King's College London.

## Acknowledgements

The authors from CASIA thank the Strategic Priority Research Program of Chinese Academy of Sciences, Grant No. XDA27030401.

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
