# Appendices

# A Preliminary Remarks

**Remark 1.** *The multi-agent state-action value function obeys the bounds*

$$\left| Q_{\boldsymbol{\theta}}^{i_1,\dots,i_k} \left( s, \boldsymbol{a}^{(i_1,\dots,i_k)} \right) \right| \leq \frac{\beta}{1-\gamma}, \quad \text{for all } s \in \mathcal{S}, \; \boldsymbol{a}^{(i_1,\dots,i_k)} \in \mathcal{A}^{(i_1,\dots,i_k)}.$$

*Proof.* It suffices to prove that, for all $t$, the total reward satisfies $|R_t| \leq \frac{\beta}{1-\gamma}$, as the value functions are expectations of it. We have

$$|R_t| = \left| \sum_{k=0}^{\infty} \gamma^k \mathrm{r}_{t+k} \right| \leq \sum_{k=0}^{\infty} |\gamma^k \mathrm{r}_{t+k}| \leq \sum_{k=0}^{\infty} \gamma^k \beta = \frac{\beta}{1-\gamma}$$

$\square$

**Remark 2.** *The multi-agent advantage function is bounded.*

*Proof.* We have

$$\left| A_{\boldsymbol{\theta}}^{i_1,\dots,i_k} \left( s, \boldsymbol{a}^{(j_1,\dots,j_m)}, \boldsymbol{a}^{(i_1,\dots,i_k)} \right) \right|$$
$$= \left| Q_{\boldsymbol{\theta}}^{j_1,\dots,j_m,i_1,\dots,i_k} \left( s, \boldsymbol{a}^{(j_1,\dots,j_m,i_1,\dots,i_k)} \right) - Q_{\boldsymbol{\theta}}^{j_1,\dots,j_m} \left( s, \boldsymbol{a}^{(j_1,\dots,j_m)} \right) \right|$$
$$\leq \left| Q_{\boldsymbol{\theta}}^{j_1,\dots,j_m,i_1,\dots,i_k} \left( s, \boldsymbol{a}^{(j_1,\dots,j_m,i_1,\dots,i_k)} \right) \right| + \left| Q_{\boldsymbol{\theta}}^{j_1,\dots,j_m} \left( s, \boldsymbol{a}^{(j_1,\dots,j_m)} \right) \right| \leq \frac{2\beta}{1-\gamma}$$

$\square$

**Remark 3.** *Baselines in MARL have the following property*

$$\mathbb{E}_{\mathrm{s} \sim d_{\boldsymbol{\theta}}^t, \mathbf{a} \sim \boldsymbol{\pi}_{\boldsymbol{\theta}}} \left[ b \left( \mathrm{s}, \mathbf{a}^{-i} \right) \nabla_{\theta^i} \log \pi_{\boldsymbol{\theta}}^i (\mathrm{a}^i | \mathrm{s}) \right] = \mathbf{0}.$$

*Proof.* We have

$$\mathbb{E}_{\mathrm{s} \sim d_{\boldsymbol{\theta}}^t, \mathbf{a} \sim \boldsymbol{\pi}_{\boldsymbol{\theta}}} \left[ b \left( \mathrm{s}, \mathbf{a}^{-i} \right) \nabla_{\theta^i} \log \pi_{\boldsymbol{\theta}}^i (\mathrm{a}^i | \mathrm{s}) \right] = \mathbb{E}_{\mathrm{s} \sim d_{\boldsymbol{\theta}}^t, \mathbf{a}^{-i} \sim \boldsymbol{\pi}_{\boldsymbol{\theta}}^{-i}} \left[ b \left( \mathrm{s}, \mathbf{a}^{-i} \right) \mathbb{E}_{\mathrm{a}^i \sim \pi_{\boldsymbol{\theta}}^i} \left[ \nabla_{\theta^i} \log \pi_{\boldsymbol{\theta}}^i (\mathrm{a}^i | \mathrm{s}) \right] \right],$$

which means that it suffices to prove that for any $s \in \mathcal{S}$

$$\mathbb{E}_{\mathrm{a}^i \sim \pi_{\boldsymbol{\theta}}^i} \left[ \nabla_{\theta^i} \log \pi_{\boldsymbol{\theta}}^i (\mathrm{a}^i | \mathrm{s}) \right] = \mathbf{0}.$$

We prove it for continuous $\mathcal{A}^i$. The discrete case is analogous.

$$\mathbb{E}_{\mathrm{a}^i \sim \pi_{\boldsymbol{\theta}}^i} \left[ \nabla_{\theta^i} \log \pi_{\boldsymbol{\theta}}^i (\mathrm{a}^i | s) \right] = \int_{\mathcal{A}^i} \pi_{\boldsymbol{\theta}}^i \left( a^i | s \right) \nabla_{\theta^i} \log \pi_{\boldsymbol{\theta}}^i (\mathrm{a}^i | s) \, da^i$$
$$= \int_{\mathcal{A}^i} \nabla_{\theta^i} \pi_{\boldsymbol{\theta}}^i \left( a^i | s \right) \, da^i = \nabla_{\theta^i} \int_{\mathcal{A}^i} \pi_{\boldsymbol{\theta}}^i \left( a^i | s \right) \, da^i = \nabla_{\theta^i} (1) = \mathbf{0}$$

$\square$

# B Proofs of the Theoretical Results

## B.1 Proofs of Lemmas 1, 2, and 3

In this subsection, we prove the lemmas stated in the paper. We realise that their application to other, very complex, proofs is not always immediately clear. To compensate for that, we provide the stronger versions of the lemmas; we give a detailed proof of the strong version of Lemma 1 which is supposed to demonstrate the equivalence of the normal and strong versions, and prove the normal versions of Lemmas 2 & 3, and state their stronger versions as remarks to the proofs.

**Lemma 1** (Multi-agent advantage decomposition). *For any state $s \in \mathcal{S}$, the following equation holds for any subset of $m$ agents and any permutation of their labels,*

$$A_{\boldsymbol{\theta}}^{1,\dots,m}\left(s, \boldsymbol{a}^{(1,\dots,m)}\right) = \sum_{i=1}^{m} A_{\boldsymbol{\theta}}^{i}\left(s, \boldsymbol{a}^{(1,\dots,i-1)}, a^{i}\right).$$

*Proof.* We prove a slightly **stronger**, but perhaps less telling, version of the lemma, which is

$$A_{\boldsymbol{\theta}}^{k+1,\dots,m}\left(s, \boldsymbol{a}^{(1,\dots,k)}, \boldsymbol{a}^{(k+1,\dots,m)}\right) = \sum_{i=k+1}^{m} A_{\boldsymbol{\theta}}^{i}\left(s, \boldsymbol{a}^{(1,\dots,i-1)}, a^{i}\right). \tag{13}$$

The original form of the lemma will follow from the above by taking $k = 0$.
By the definition of the multi-agent advantage, we have

$$A_{\boldsymbol{\theta}}^{k+1,\dots,m}\left(s, \boldsymbol{a}^{(1,\dots,k)}, \boldsymbol{a}^{(k+1,\dots,m)}\right)$$
$$= Q_{\boldsymbol{\theta}}^{1,\dots,k,k+1,\dots,m}\left(s, \boldsymbol{a}^{(1,\dots,k,k+1,\dots,m)}\right) - Q_{\boldsymbol{\theta}}^{1,\dots,k}\left(s, \boldsymbol{a}^{(1,\dots,k)}\right)$$

which can be written as a telescoping sum

$$Q_{\boldsymbol{\theta}}^{1,\dots,k,k+1,\dots,m}\left(s, \boldsymbol{a}^{(1,\dots,k,k+1,\dots,m)}\right) - Q_{\boldsymbol{\theta}}^{1,\dots,k}\left(s, \boldsymbol{a}^{(1,\dots,k)}\right)$$
$$= \sum_{i=k+1}^{m} \left[Q_{\boldsymbol{\theta}}^{(1,\dots,i)}\left(s, \boldsymbol{a}^{(1,\dots,i)}\right) - Q_{\boldsymbol{\theta}}^{(1,\dots,i-1)}\left(s, \boldsymbol{a}^{(1,\dots,i-1)}\right)\right]$$
$$= \sum_{i=k+1}^{m} A_{\boldsymbol{\theta}}^{i}\left(s, \boldsymbol{a}^{(1,\dots,i-1)}, a^{i}\right)$$

$\square$

**Lemma 2.** *For any state $s \in \mathcal{S}$, we have*
$$\mathbf{Var}_{\mathbf{a} \sim \boldsymbol{\pi}_{\boldsymbol{\theta}}}\left[A_{\boldsymbol{\theta}}(s, \mathbf{a})\right] = \sum_{i=1}^{n} \mathbb{E}_{a^{1} \sim \pi_{\boldsymbol{\theta}}^{1}, \dots, a^{i-1} \sim \pi_{\boldsymbol{\theta}}^{i-1}}\left[\mathbf{Var}_{a^{i} \sim \pi_{\boldsymbol{\theta}}^{i}}\left[A_{\boldsymbol{\theta}}^{i}\left(s, \mathbf{a}^{(1,\dots,i-1)}, a^{i}\right)\right]\right].$$

*Proof.* The trick of this proof is to develop a relation on the variance of multi-agent advantage which is recursive over the number of agents. We have

$$\mathbf{Var}_{\mathbf{a} \sim \boldsymbol{\pi}_{\boldsymbol{\theta}}}\left[A_{\boldsymbol{\theta}}(s, \mathbf{a})\right] = \mathbf{Var}_{a^{1} \sim \pi_{\boldsymbol{\theta}}^{1}, \dots, a^{v} \sim \pi_{\boldsymbol{\theta}}^{v}}\left[A_{\boldsymbol{\theta}}^{1,\dots,n}\left(s, \mathbf{a}^{(1,\dots,n)}\right)\right]$$
$$= \mathbb{E}_{a^{1} \sim \pi_{\boldsymbol{\theta}}^{1}, \dots, a^{n} \sim \pi_{\boldsymbol{\theta}}^{n}}\left[A_{\boldsymbol{\theta}}^{1,\dots,n}\left(s, \mathbf{a}^{(1,\dots,n)}\right)^{2}\right]$$
$$= \mathbb{E}_{a^{1} \sim \pi_{\boldsymbol{\theta}}^{1}, \dots, a^{n-1} \sim \pi_{\boldsymbol{\theta}}^{n-1}}\left[\mathbb{E}_{a^{n} \sim \pi_{\boldsymbol{\theta}}^{n}}\left[A_{\boldsymbol{\theta}}^{1,\dots,n}\left(s, \mathbf{a}^{(1,\dots,n)}\right)^{2}\right]\right.$$
$$\left. - \mathbb{E}_{a^{n} \sim \pi_{\boldsymbol{\theta}}^{n}}\left[A_{\boldsymbol{\theta}}^{1,\dots,n}\left(s, \mathbf{a}^{(1,\dots,n)}\right)\right]^{2} + \mathbb{E}_{a^{n} \sim \pi_{\boldsymbol{\theta}}^{n}}\left[A_{\boldsymbol{\theta}}^{1,\dots,n}\left(s, \mathbf{a}^{(1,\dots,n)}\right)\right]^{2}\right]$$
$$= \mathbb{E}_{a^{1} \sim \pi_{\boldsymbol{\theta}}^{1}, \dots, a^{n-1} \sim \pi_{\boldsymbol{\theta}}^{n-1}}\left[\mathbf{Var}_{a^{n} \sim \pi_{\boldsymbol{\theta}}^{n}}\left[A_{\boldsymbol{\theta}}^{1,\dots,n}\left(s, \mathbf{a}^{(1,\dots,n)}\right)\right]\right]$$
$$+ \mathbb{E}_{a^{1} \sim \pi_{\boldsymbol{\theta}}^{1}, \dots, a^{n-1} \sim \pi_{\boldsymbol{\theta}}^{n-1}}\left[A_{\boldsymbol{\theta}}^{1,\dots,n-1}\left(s, \mathbf{a}^{(1,\dots,n-1)}\right)^{2}\right]$$

which, by the stronger version of Lemma 1, given by Equation 13, applied to the first term, equals

$$\mathbb{E}_{a^1 \sim \pi_{\boldsymbol{\theta}}^1, \ldots, a^{n-1} \sim \pi_{\boldsymbol{\theta}}^{n-1}} \left[ \mathbf{Var}_{a^n \sim \pi_{\boldsymbol{\theta}}^n} \left[ A_{\boldsymbol{\theta}}^n \left( s, \mathbf{a}^{(1,\ldots,n-1)}, a^n \right) \right] \right]$$

$$+ \mathbb{E}_{a^1 \sim \pi_{\boldsymbol{\theta}}^1, \ldots, a^{n-1} \sim \pi_{\boldsymbol{\theta}}^{n-1}} \left[ A_{\boldsymbol{\theta}}^{1,\ldots,n-1} \left( s, \mathbf{a}^{(1,\ldots,n-1)} \right)^2 \right]$$

Hence, we have a recursive relation

$$\mathbf{Var}_{a^1 \sim \pi_{\boldsymbol{\theta}}^1, \ldots, a^v \sim \pi_{\boldsymbol{\theta}}^v} \left[ A_{\boldsymbol{\theta}}^{1,\ldots,n} \left( s, \mathbf{a}^{(1,\ldots,n)} \right) \right]$$

$$= \mathbb{E}_{a^1 \sim \pi_{\boldsymbol{\theta}}^1, \ldots, a^{n-1} \sim \pi_{\boldsymbol{\theta}}^{n-1}} \left[ \mathbf{Var}_{a^n \sim \pi_{\boldsymbol{\theta}}^n} \left[ A_{\boldsymbol{\theta}}^n \left( s, \mathbf{a}^{(1,\ldots,n-1)}, a^n \right) \right] \right]$$

$$+ \mathbf{Var}_{a^1 \sim \pi_{\boldsymbol{\theta}}^1, \ldots, a^{n-1} \sim \pi_{\boldsymbol{\theta}}^{n-1}} \left[ A_{\boldsymbol{\theta}}^{1,\ldots,n-1} \left( s, \mathbf{a}^{(1,\ldots,n-1)} \right) \right]$$

from which we can obtain

$$\mathbf{Var}_{a^1 \sim \pi_{\boldsymbol{\theta}}^1, \ldots, a^v \sim \pi_{\boldsymbol{\theta}}^v} \left[ A_{\boldsymbol{\theta}}^{1,\ldots,n} \left( s, \mathbf{a}^{(1,\ldots,n)} \right) \right]$$

$$= \sum_{i=1}^{n} \mathbb{E}_{a^1 \sim \pi_{\boldsymbol{\theta}}^1, \ldots, a^{i-1} \sim \pi_{\boldsymbol{\theta}}^{i-1}} \left[ \mathbf{Var}_{a^i \sim \pi_{\boldsymbol{\theta}}^i} \left[ A_{\boldsymbol{\theta}}^i \left( s, \mathbf{a}^{(1,\ldots,i-1)}, a^i \right) \right] \right]$$

$\square$

**Remark 4.** *Lemma 2 has a **stronger** version, coming as a corollary to the above proof; that is*

$$\mathbf{Var}_{a^{k+1} \sim \pi_{\boldsymbol{\theta}}^{k+1}, \ldots, a^n \sim \pi_{\boldsymbol{\theta}}^n} \left[ A^{k+1,\ldots,n} \left( s, \boldsymbol{a}^{(1,\ldots,k)}, \mathbf{a}^{(k+1,\ldots,n)} \right) \right]$$

$$= \sum_{i=k+1}^{n} \mathbb{E}_{a^{k+1} \sim \pi_{\boldsymbol{\theta}}^{k+1}, \ldots, a^{i-1} \sim \pi_{\boldsymbol{\theta}}^{i-1}} \left[ \mathbf{Var}_{a^i \sim \pi_{\boldsymbol{\theta}}^i} \left[ A_{\boldsymbol{\theta}}^i \left( s, \boldsymbol{a}^{1,\ldots,k}, \mathbf{a}^{k+1,\ldots,i-1}, a^i \right) \right] \right]. \quad (14)$$

*We think of it as a corollary to the proof of the lemma, as the fixed joint action $\boldsymbol{a}^{1,\ldots,k}$ has the same algebraic properites, throughout the proof, as state $s$.*

**Lemma 3.** *For any state $s \in \mathcal{S}$, we have*

$$\mathbf{Var}_{\mathbf{a} \sim \boldsymbol{\pi}_{\boldsymbol{\theta}}} \left[ A_{\boldsymbol{\theta}}(s, \mathbf{a}) \right] \leq \sum_{i=1}^{n} \mathbf{Var}_{\mathbf{a}^{-i} \sim \boldsymbol{\pi}_{\boldsymbol{\theta}}^{-i}, a^i \sim \pi_{\boldsymbol{\theta}}^i} \left[ A_{\boldsymbol{\theta}}^i(s, \mathbf{a}^{-i}, a^i) \right].$$

*Proof.* By Lemma 2, we have

$$\mathbf{Var}_{\mathbf{a} \sim \boldsymbol{\pi}_{\boldsymbol{\theta}}} \left[ A_{\boldsymbol{\theta}}(s, \mathbf{a}) \right] = \sum_{i=1}^{n} \mathbb{E}_{a^1 \sim \pi_{\boldsymbol{\theta}}^1, \ldots, a^{i-1} \sim \pi_{\boldsymbol{\theta}}^{i-1}} \left[ \mathbf{Var}_{a^i \sim \pi_{\boldsymbol{\theta}}^i} \left[ A_{\boldsymbol{\theta}}^i \left( s, \mathbf{a}^{(1,\ldots,i-1)}, a^i \right) \right] \right] \quad (15)$$

Take an arbitrary $i$. We have

$$\mathbb{E}_{a^1 \sim \pi_{\boldsymbol{\theta}}^1, \ldots, a^{i-1} \sim \pi_{\boldsymbol{\theta}}^{i-1}} \left[ \mathbf{Var}_{a^i \sim \pi_{\boldsymbol{\theta}}^i} \left[ A_{\boldsymbol{\theta}}^i \left( s, \mathbf{a}^{(1,\ldots,i-1)}, a^i \right) \right] \right]$$

$$= \mathbb{E}_{a^1 \sim \pi_{\boldsymbol{\theta}}^1, \ldots, a^{i-1} \sim \pi_{\boldsymbol{\theta}}^{i-1}} \left[ \mathbb{E}_{a^i \sim \pi_{\boldsymbol{\theta}}^i} \left[ A_{\boldsymbol{\theta}}^i \left( s, \mathbf{a}^{(1,\ldots,i-1)}, a^i \right)^2 \right] \right]$$

$$= \mathbb{E}_{a^1 \sim \pi_{\boldsymbol{\theta}}^1, \ldots, a^{i-1} \sim \pi_{\boldsymbol{\theta}}^{i-1}} \left[ \mathbb{E}_{a^i \sim \pi_{\boldsymbol{\theta}}^i} \left[ \mathbb{E}_{a^{i+1} \sim \pi_{\boldsymbol{\theta}}^{i+1}, \ldots, a^n \sim \pi_{\boldsymbol{\theta}}^n} \left[ A_{\boldsymbol{\theta}}^{i,\ldots,n} \left( s, \mathbf{a}^{(1,\ldots,i-1)}, \mathbf{a}^{(i,\ldots,n)} \right) \right]^2 \right] \right]$$

$$\leq \mathbb{E}_{a^1 \sim \pi_{\boldsymbol{\theta}}^1, \ldots, a^{i-1} \sim \pi_{\boldsymbol{\theta}}^{i-1}} \left[ \mathbb{E}_{a^i \sim \pi_{\boldsymbol{\theta}}^i} \left[ \mathbb{E}_{a^{i+1} \sim \pi_{\boldsymbol{\theta}}^{i+1}, \ldots, a^n \sim \pi_{\boldsymbol{\theta}}^n} \left[ A_{\boldsymbol{\theta}}^{i,\ldots,n} \left( s, \mathbf{a}^{(1,\ldots,i-1)}, \mathbf{a}^{(i,\ldots,n)} \right)^2 \right] \right] \right]$$

$$= \mathbb{E}_{a^1 \sim \pi_{\boldsymbol{\theta}}^1, \ldots, a^{i-1} \sim \pi_{\boldsymbol{\theta}}^{i-1}, a^{i+1} \sim \pi_{\boldsymbol{\theta}}^{i+1}, \ldots, a^n \sim \pi_{\boldsymbol{\theta}}^n} \left[ \mathbb{E}_{a^i \sim \pi_{\boldsymbol{\theta}}^i} \left[ A_{\boldsymbol{\theta}}^{i,\ldots,n} \left( s, \mathbf{a}^{(1,\ldots,i-1)}, \mathbf{a}^{(i,\ldots,n)} \right)^2 \right] \right]$$

The above can be equivalently, but more tellingly, rewritten after permuting (cyclic shift) the labels of agents, in the following way

$$\mathbb{E}_{\mathbf{a}^{-i} \sim \boldsymbol{\pi}_{\boldsymbol{\theta}}^{-i}} \left[ \mathbb{E}_{a^i \sim \pi_{\boldsymbol{\theta}}^i} \left[ A_{\boldsymbol{\theta}}^{i+1,\ldots,n,i} \left( s, \mathbf{a}^{(1,\ldots,i-1)}, \mathbf{a}^{(i+1,\ldots,n,i)} \right)^2 \right] \right]$$

$$= \mathbb{E}_{\mathbf{a}^{-i} \sim \boldsymbol{\pi}_{\boldsymbol{\theta}}^{-i}} \left[ \mathbf{Var}_{a^i \sim \pi_{\boldsymbol{\theta}}^i} \left[ A_{\boldsymbol{\theta}}^{i+1,\ldots,n,i} \left( s, \mathbf{a}^{(1,\ldots,i-1)}, \mathbf{a}^{(i+1,\ldots,n,i)} \right) \right] \right]$$

which, by the strong version of Lemma 1, equals

$$\mathbb{E}_{\mathbf{a}^{-i} \sim \pi_{\boldsymbol{\theta}}^{-i}} \left[ \mathbf{Var}_{\mathrm{a}^i \sim \pi_{\boldsymbol{\theta}}^i} \left[ A_{\boldsymbol{\theta}}^i \left( s, \mathbf{a}^{-i}, \mathrm{a}^i \right) \right] \right]$$

which can be further simplified by

$$\mathbb{E}_{\mathbf{a}^{-i} \sim \pi_{\boldsymbol{\theta}}^{-i}} \left[ \mathbf{Var}_{\mathrm{a}^i \sim \pi_{\boldsymbol{\theta}}^i} \left[ A_{\boldsymbol{\theta}}^i \left( s, \mathbf{a}^{-i}, \mathrm{a}^i \right) \right] \right] = \mathbb{E}_{\mathbf{a}^{-i} \sim \pi_{\boldsymbol{\theta}}^{-i}} \left[ \mathbb{E}_{\mathrm{a}^i \sim \pi_{\boldsymbol{\theta}}^i} \left[ A_{\boldsymbol{\theta}}^i \left( s, \mathbf{a}^{-i}, \mathrm{a}^i \right)^2 \right] \right]$$

$$= \mathbb{E}_{\mathbf{a} \sim \boldsymbol{\pi}_{\boldsymbol{\theta}}} \left[ A_{\boldsymbol{\theta}}^i \left( s, \mathbf{a}^{-i}, \mathrm{a}^i \right)^2 \right] = \mathbf{Var}_{\mathbf{a} \sim \boldsymbol{\pi}_{\boldsymbol{\theta}}} \left[ A_{\boldsymbol{\theta}}^i \left( s, \mathbf{a}^{-i}, \mathrm{a}^i \right) \right]$$

which, combined with Equation 15, finishes the proof. □

**Remark 5.** *Again, subsuming a joint action $\mathbf{a}^{(1,\ldots,k)}$ into state in the above proof, we can have a **stronger** version of Lemma 3,*

$$\mathbf{Var}_{\mathrm{a}^{k+1} \sim \pi_{\boldsymbol{\theta}}^{k+1}, \ldots, \mathrm{a}^n \sim \pi_{\boldsymbol{\theta}}^n} \left[ A_{\boldsymbol{\theta}}^{k+1,\ldots,n} \left( s, \boldsymbol{a}^{(1,\ldots,k)}, \mathbf{a}^{(k+1,\ldots,n)} \right) \right]$$

$$\leq \sum_{i=k+1}^n \mathbf{Var}_{\mathrm{a}^{k+1} \sim \pi_{\boldsymbol{\theta}}^{k+1}, \ldots, \mathrm{a}^n \sim \pi_{\boldsymbol{\theta}}^n} \left[ A^i \left( s, \boldsymbol{a}^{(k+1,\ldots,i-1,i+1,\ldots,n)}, \mathrm{a}^i \right) \right] \tag{16}$$

## B.2 Proofs of Theorems 1 and 2

Let us recall the two assumptions that we make in the paper.

**Assumption 1.** *The state space $\mathcal{S}$, and every agent $i$'s action space $\mathcal{A}^i$ is either discrete and finite, or continuous and compact.*

**Assumption 2.** *For all $i \in \mathcal{N}$, $s \in \mathcal{S}$, $a^i \in \mathcal{A}^i$, the map $\theta^i \mapsto \pi_{\boldsymbol{\theta}}^i(a^i|s)$ is continuously differentiable.*

These assumptions assure that the supremum $\sup_{s,a^i} \left\| \nabla_{\theta^i} \log \pi_{\boldsymbol{\theta}}^i(a^i|s) \right\|$ exists for every agent $i$. We notice that the supremum $\sup_{s,\boldsymbol{a}^{-i},a^i} \left| A^i(s, \boldsymbol{a}^{-i}, a^i) \right|$ exists regardless of assumptions, as by Remark 2, the multi-agent advantage is bounded from both sides.

**Theorem 1.** *The CTDE and DT estimators of MAPG satisfy*

$$\mathbf{Var}_{\mathrm{s}_{0:\infty} \sim d_{\boldsymbol{\theta}}^{0:\infty}, \mathbf{a}_{0:\infty} \sim \boldsymbol{\pi}_{\boldsymbol{\theta}}} \left[ \mathbf{g}_C^i \right] - \mathbf{Var}_{\mathrm{s}_{0:\infty} \sim d_{\boldsymbol{\theta}}^{0:\infty}, \mathbf{a}_{0:\infty} \sim \boldsymbol{\pi}_{\boldsymbol{\theta}}} \left[ \mathbf{g}_D^i \right] \ \leq \ \frac{B_i^2}{1-\gamma^2} \sum_{j \neq i} \epsilon_j^2 \ \leq \ (n-1) \frac{(\epsilon B_i)^2}{1-\gamma^2}$$

*where $B_i = \sup_{s,\boldsymbol{a}} \left\| \nabla_{\theta^i} \log \pi_{\boldsymbol{\theta}}^i \left( \mathrm{a}^i|s \right) \right\|$, $\epsilon_i = \sup_{s,\boldsymbol{a}^{-i},a^i} \left| A_{\boldsymbol{\theta}}^i(s, \boldsymbol{a}^{-i}, a^i) \right|$, and $\epsilon = \max_i \epsilon_i$.*

*Proof.* It suffices to prove the first inequality, as the second one is a trivial upper bound. Let's consider an arbitrary time step $t \geq 0$. Let

$$\mathbf{g}_{C,t}^i = \hat{Q}(\mathrm{s}_t, \mathbf{a}_t) \nabla_{\theta^i} \log \pi_{\boldsymbol{\theta}}^i(\mathrm{s}_t, \mathrm{a}_t^i)$$

$$\mathbf{g}_{D,t}^i = \hat{Q}^i(\mathrm{s}_t, \mathrm{a}_t^i) \nabla_{\theta^i} \log \pi_{\boldsymbol{\theta}}^i(\mathrm{s}_t, \mathrm{a}_t^i)$$

be the contributions to the centralised and decentralised gradient estimators coming from sampling $\mathrm{s}_t \sim d_{\boldsymbol{\theta}}^t$, $\mathbf{a}_t \sim \boldsymbol{\pi}_{\boldsymbol{\theta}}$. Note that

$$\mathbf{g}_C^i = \sum_{t=0}^{\infty} \gamma^t \mathbf{g}_{C,t}^i \qquad \text{and} \qquad \mathbf{g}_D^i = \sum_{t=0}^{\infty} \gamma^t \mathbf{g}_{D,t}^i$$

Moreover, let $\mathbf{g}_{C,t,j}^i$ and $\mathbf{g}_{D,t,j}^i$ be the $j^{th}$ components of $\mathbf{g}_{C,t}^i$ and $\mathbf{g}_{D,t}^i$, respectively. Using the law of total variance, we have

$$\mathbf{Var}_{s \sim d_{\boldsymbol{\theta}}^t, \mathbf{a} \sim \boldsymbol{\pi}_{\boldsymbol{\theta}}} \left[ \mathbf{g}_{C,t,j}^i \right] - \mathbf{Var}_{s \sim d_{\boldsymbol{\theta}}^t, \mathbf{a} \sim \boldsymbol{\pi}_{\boldsymbol{\theta}}} \left[ \mathbf{g}_{D,t,j}^i \right]$$

$$= \left( \mathbf{Var}_{s \sim d_{\boldsymbol{\theta}}^t} \left[ \mathbb{E}_{\mathbf{a} \sim \boldsymbol{\pi}_{\boldsymbol{\theta}}} \left[ \mathbf{g}_{C,t,j}^i \right] \right] + \mathbb{E}_{s \sim d_{\boldsymbol{\theta}}^t} \left[ \mathbf{Var}_{\mathbf{a} \sim \boldsymbol{\pi}_{\boldsymbol{\theta}}} \left[ \mathbf{g}_{C,t,j}^i \right] \right] \right)$$

$$- \left( \mathbf{Var}_{s \sim d_{\boldsymbol{\theta}}^t} \left[ \mathbb{E}_{\mathbf{a} \sim \boldsymbol{\pi}_{\boldsymbol{\theta}}} \left[ \mathbf{g}_{D,t,j}^i \right] \right] + \mathbb{E}_{s \sim d_{\boldsymbol{\theta}}^t} \left[ \mathbf{Var}_{\mathbf{a} \sim \boldsymbol{\pi}_{\boldsymbol{\theta}}} \left[ \mathbf{g}_{D,t,j}^i \right] \right] \right) \tag{17}$$

Noting that $\mathbf{g}_{\mathrm{C}}^i$ and $\mathbf{g}_{\mathrm{D}}^i$ have the same expectation over $\mathbf{a} \sim \boldsymbol{\pi_\theta}$, the above simplifies to

$$\mathbb{E}_{s \sim d_{\boldsymbol{\theta}}^t} \left[ \mathbf{Var}_{\mathbf{a} \sim \boldsymbol{\pi_\theta}} \left[ \mathbf{g}_{\mathrm{C},t,j}^i \right] \right] - \mathbb{E}_{s \sim d_{\boldsymbol{\theta}}^t} \left[ \mathbf{Var}_{\mathbf{a} \sim pi_{\boldsymbol{\theta}}} \left[ \mathbf{g}_{\mathrm{D},t,j}^i \right] \right]$$
$$= \mathbb{E}_{s \sim d_{\boldsymbol{\theta}}^t} \left[ \mathbf{Var}_{\mathbf{a} \sim \boldsymbol{\pi_\theta}} \left[ \mathbf{g}_{\mathrm{C},t,j}^i \right] - \mathbf{Var}_{\mathbf{a} \sim \boldsymbol{\pi_\theta}} \left[ \mathbf{g}_{\mathrm{D},t,j}^i \right] \right] \tag{18}$$

Let's fix a state $s$. Using (again) the fact that the expectations of the two gradients are the same, we have

$$\mathbf{Var}_{\mathbf{a} \sim \boldsymbol{\pi_\theta}} \left[ \mathbf{g}_{\mathrm{C},t,j}^i \right] - \mathbf{Var}_{\mathbf{a} \sim \boldsymbol{\pi_\theta}} \left[ \mathbf{g}_{\mathrm{D},t,j}^i \right]$$
$$= \left( \mathbb{E}_{\mathbf{a} \sim \boldsymbol{\pi_\theta}} \left[ \left( \mathbf{g}_{\mathrm{C},t,j}^i \right)^2 \right] - \mathbb{E}_{\mathbf{a} \sim \boldsymbol{\pi_\theta}} \left[ \mathbf{g}_{\mathrm{C},t,j}^i \right]^2 \right) - \left( \mathbb{E}_{\mathbf{a} \sim \boldsymbol{\pi_\theta}} \left[ \left( \mathbf{g}_{\mathrm{D},t,j}^i \right)^2 \right] - \mathbb{E}_{\mathbf{a} \sim \boldsymbol{\pi_\theta}} \left[ \mathbf{g}_{\mathrm{D},t,j}^i \right]^2 \right)$$
$$= \mathbb{E}_{\mathbf{a} \sim \boldsymbol{\pi_\theta}} \left[ \left( \mathbf{g}_{\mathrm{C},t,j}^i \right)^2 \right] - \mathbb{E}_{\mathbf{a} \sim \boldsymbol{\pi_\theta}} \left[ \left( \mathbf{g}_{\mathrm{D},t,j}^i \right)^2 \right]$$
$$= \mathbb{E}_{\mathbf{a} \sim \boldsymbol{\pi_\theta}} \left[ \left( \mathbf{g}_{\mathrm{C},t,j}^i \right)^2 - \left( \mathbf{g}_{\mathrm{D},t,j}^i \right)^2 \right]$$
$$= \mathbb{E}_{\mathbf{a} \sim \boldsymbol{\pi_\theta}} \left[ \left( \frac{\partial \log \pi_{\boldsymbol{\theta}}^i(\mathbf{a}^i|s)}{\partial \theta^i} \hat{Q}(s, \mathbf{a}) \right)^2 - \left( \frac{\partial \log \pi_{\boldsymbol{\theta}}^i(\mathbf{a}^i|s)}{\partial \theta^i} \hat{Q}^i(s, \mathbf{a}^i) \right)^2 \right]$$
$$= \mathbb{E}_{\mathbf{a}^i \sim \pi_{\boldsymbol{\theta}}^i} \left[ \left( \frac{\partial \log \pi_{\boldsymbol{\theta}}^i(\mathbf{a}^i|s)}{\partial \theta^i} \right)^2 \mathbb{E}_{\mathbf{a}^{-i} \sim \boldsymbol{\pi}_{\boldsymbol{\theta}}^{-i}} \left[ \hat{Q}(s, \mathbf{a}^i, \mathbf{a}^{-i})^2 - \hat{Q}^i(s, \mathbf{a}^i)^2 \right] \right].$$

The inner expectation is the variance of $\hat{Q}(s, \mathbf{a}^i, \mathbf{a}^{-i})$, given $\mathbf{a}^i$. We rewrite it as

$$= \mathbb{E}_{\mathbf{a}^i \sim \pi_{\boldsymbol{\theta}}^i} \left[ \left( \frac{\partial \log \pi_{\boldsymbol{\theta}}^i(\mathbf{a}^i|s)}{\partial \theta^i} \right)^2 \mathbb{E}_{\mathbf{a}^{-i} \sim \boldsymbol{\pi}_{\boldsymbol{\theta}}^{-i}} \left[ \left( \hat{Q}(s, \mathbf{a}^i, \mathbf{a}^{-i}) - \hat{Q}^i(s, \mathbf{a}^i) \right)^2 \right] \right]$$
$$= \mathbb{E}_{\mathbf{a} \sim \boldsymbol{\pi_\theta}} \left[ \left( \frac{\partial \log \pi_{\boldsymbol{\theta}}^i(\mathbf{a}^i|s)}{\partial \theta^i} \right)^2 \left( \hat{Q}(s, \mathbf{a}) - \hat{Q}^i(s, \mathbf{a}^i) \right)^2 \right].$$

Now, recalling that the variance of the total gradient is the sum of variances of the gradient components, we have

$$\mathbf{Var}_{\mathbf{a} \sim \boldsymbol{\pi_\theta}} \left[ \mathbf{g}_{\mathrm{C},t}^i \right] - \mathbf{Var}_{\mathbf{a} \sim \boldsymbol{\pi_\theta}} \left[ \mathbf{g}_{\mathrm{D},t}^i \right] = \mathbb{E}_{\mathbf{a} \sim \boldsymbol{\pi_\theta}} \left[ \left|\left| \nabla_{\theta^i} \log \pi_{\boldsymbol{\theta}}^i(\mathbf{a}^i|s) \right|\right|^2 \left( \hat{Q}(s, \mathbf{a}) - \hat{Q}^i(s, \mathbf{a}^i) \right)^2 \right]$$
$$\leq B_i^2 \, \mathbb{E}_{\mathbf{a} \sim \boldsymbol{\pi_\theta}} \left[ \left( \hat{Q}(s, \mathbf{a}) - \hat{Q}^i(s, \mathbf{a}^i) \right)^2 \right] = B_i^2 \, \mathbb{E}_{\mathbf{a}^i \sim \pi_{\boldsymbol{\theta}}^i} \left[ \mathbb{E}_{\mathbf{a}^{-i} \sim \boldsymbol{\pi}_{\boldsymbol{\theta}}^{-i}} \left[ \left( \hat{Q}(s, \mathbf{a}) - \hat{Q}^i(s, \mathbf{a}^i) \right)^2 \right] \right]$$
$$= B_i^2 \, \mathbb{E}_{\mathbf{a}^i \sim \pi_{\boldsymbol{\theta}}^i} \left[ \mathbb{E}_{\mathbf{a}^{-i} \sim \boldsymbol{\pi}_{\boldsymbol{\theta}}^{-i}} \left[ \left( \hat{Q}(s, \mathbf{a}^i, \mathbf{a}^{-i}) - \hat{Q}^i(s, \mathbf{a}^i) \right)^2 \right] \right]$$
$$= B_i^2 \, \mathbb{E}_{\mathbf{a}^i \sim \pi_{\boldsymbol{\theta}}^i} \left[ \mathbb{E}_{\mathbf{a}^{-i} \sim \boldsymbol{\pi}_{\boldsymbol{\theta}}^{-i}} \left[ \hat{A}^{-i}(s, \mathbf{a}^i, \mathbf{a}^{-i})^2 \right] \right] = B_i^2 \, \mathbb{E}_{\mathbf{a}^i \sim \pi_{\boldsymbol{\theta}}^i} \left[ \mathbf{Var}_{\mathbf{a}^{-i} \sim \boldsymbol{\pi}_{\boldsymbol{\theta}}^{-i}} \left[ \hat{A}^{-i}(s, \mathbf{a}^i, \mathbf{a}^{-i}) \right] \right]$$

which by the strong version of Lemma 3, given in Equation 16, can be upper-bounded by

$$B_i^2 \, \mathbb{E}_{\mathbf{a}^i \sim \pi_{\boldsymbol{\theta}}^i} \left[ \sum_{j \neq i} \mathbf{Var}_{\mathbf{a}^{-i} \sim \boldsymbol{\pi}_{\boldsymbol{\theta}}^{-i}} \left[ \hat{A}^j(s, \mathbf{a}^{-j}, \mathbf{a}^j) \right] \right] = B_i^2 \sum_{j \neq i} \mathbb{E}_{\mathbf{a}^i \sim \pi_{\boldsymbol{\theta}}^i} \left[ \mathbf{Var}_{\mathbf{a}^{-i} \sim \boldsymbol{\pi}_{\boldsymbol{\theta}}^{-i}} \left[ \hat{A}^j(s, \mathbf{a}^{-j}, \mathbf{a}^j) \right] \right]$$

Notice that, for any $j \neq i$, we have

$$\mathbb{E}_{\mathbf{a}^i \sim \pi_{\boldsymbol{\theta}}^i} \left[ \mathbf{Var}_{\mathbf{a}^{-i} \sim \boldsymbol{\pi}_{\boldsymbol{\theta}}^{-i}} \left[ \hat{A}^j(s, \mathbf{a}^{-j}, \mathbf{a}^j) \right] \right]$$
$$= \mathbb{E}_{\mathbf{a}^i \sim \pi_{\boldsymbol{\theta}}^i} \left[ \mathbb{E}_{\mathbf{a}^{-i} \sim \boldsymbol{\pi}_{\boldsymbol{\theta}}^{-i}} \left[ \hat{A}^j(s, \mathbf{a}^{-j}, \mathbf{a}^j)^2 \right] \right]$$
$$= \mathbb{E}_{\mathbf{a} \sim \boldsymbol{\pi_\theta}} \left[ \hat{A}^j(s, \mathbf{a}^{-j}, \mathbf{a}^j)^2 \right] \leq \epsilon_j^2$$

This gives

$$\mathbf{Var}_{\mathbf{a} \sim \boldsymbol{\pi_\theta}} \left[ \mathbf{g}_{\mathrm{C},t}^i \right] - \mathbf{Var}_{\mathbf{a} \sim \boldsymbol{\pi_\theta}} \left[ \mathbf{g}_{\mathrm{D},t}^i \right] \leq B_i^2 \sum_{j \neq i} \epsilon_j^2$$

and combining it with Equations 17 and 18 for entire gradient vectors, we get

$$\mathbf{Var}_{s \sim d_{\boldsymbol{\theta}}^t, \mathbf{a} \sim \boldsymbol{\pi}_{\boldsymbol{\theta}}} \left[ \mathbf{g}_{\mathrm{C},t}^i \right] - \mathbf{Var}_{s \sim d_{\boldsymbol{\theta}}^t, \mathbf{a} \sim \boldsymbol{\pi}_{\boldsymbol{\theta}}} \left[ \mathbf{g}_{\mathrm{D},t}^i \right] \ \leq \ B_i^2 \sum_{j \neq i} \epsilon_j^2 \tag{19}$$

Noting that

$$\mathbf{Var}_{s_{0:\infty} \sim d_{\boldsymbol{\theta}}^{0:\infty}, \mathbf{a}_{0:\infty} \sim \boldsymbol{\pi}_{\boldsymbol{\theta}}} \left[ \mathbf{g}_{\cdot}^i \right] = \mathbf{Var}_{s_{0:\infty} \sim d_{\boldsymbol{\theta}}^{0:\infty}, \mathbf{a}_{0:\infty} \sim \boldsymbol{\pi}_{\boldsymbol{\theta}}} \left[ \sum_{t=0}^{\infty} \gamma^t \mathbf{g}_{\cdot,t}^i \right]$$

$$= \sum_{t=0}^{\infty} \mathbf{Var}_{s_t \sim d_{\boldsymbol{\theta}}^t, \mathbf{a} \sim \boldsymbol{\pi}_{\boldsymbol{\theta}}} \left[ \gamma^t \mathbf{g}_{\cdot,t}^i \right] = \sum_{t=0}^{\infty} \gamma^{2t} \mathbf{Var}_{s_t \sim d_{\boldsymbol{\theta}}^t, \mathbf{a} \sim \boldsymbol{\pi}_{\boldsymbol{\theta}}} \left[ \mathbf{g}_{\cdot,t}^i \right]$$

Combining this series expansion with the estimate from Equation 19, we finally obtain

$$\mathbf{Var}_{s_{0:\infty} \sim d_{\boldsymbol{\theta}}^{0:\infty}, \mathbf{a}_{0:\infty} \sim \boldsymbol{\pi}_{\boldsymbol{\theta}}} \left[ \mathbf{g}_{\mathrm{C}}^i \right] - \mathbf{Var}_{s_{0:\infty} \sim d_{\boldsymbol{\theta}}^{0:\infty}, \mathbf{a}_{0:\infty} \sim \boldsymbol{\pi}_{\boldsymbol{\theta}}} \left[ \mathbf{g}_{\mathrm{D}}^i \right]$$

$$\leq \ \sum_{t=0}^{\infty} \gamma^{2t} \left( B_i^2 \sum_{j \neq i} \epsilon_j^2 \right) \leq \frac{B_i^2}{1 - \gamma^2} \sum_{j \neq i} \epsilon_j^2$$

$\square$

**Theorem 2.** *The COMA and DT estimators of MAPG satisfy*

$$\mathbf{Var}_{s_{0:\infty} \sim d_{\boldsymbol{\theta}}^{0:\infty}, \mathbf{a}_{0:\infty} \sim \boldsymbol{\pi}_{\boldsymbol{\theta}}} \left[ \mathbf{g}_{COMA}^i \right] - \mathbf{Var}_{s_{0:\infty} \sim d_{\boldsymbol{\theta}}^{0:\infty}, \mathbf{a}_{0:\infty} \sim \boldsymbol{\pi}_{\boldsymbol{\theta}}} \left[ \mathbf{g}_{D}^i \right] \ \leq \ \frac{(\epsilon_i B_i)^2}{1 - \gamma^2}$$

*Proof.* Just like in the proof of Theorem 1, we start with the difference

$$\mathbf{Var}_{s \sim d_{\boldsymbol{\theta}}^t, \mathbf{a} \sim \boldsymbol{\pi}_{\boldsymbol{\theta}}} \left[ \mathbf{g}_{COMA,t,j}^i \right] - \mathbf{Var}_{s \sim d_{\boldsymbol{\theta}}^t, \mathbf{a} \sim \boldsymbol{\pi}_{\boldsymbol{\theta}}} \left[ \mathbf{g}_{D,t,j}^i \right]$$

which we transform to an analogue of Equation 18:

$$\mathbb{E}_{s \sim d_{\boldsymbol{\theta}}^t} \left[ \mathbf{Var}_{\mathbf{a} \sim \boldsymbol{\pi}_{\boldsymbol{\theta}}} \left[ \mathbf{g}_{COMA,t,j}^i \right] - \mathbf{Var}_{\mathbf{a} \sim \boldsymbol{\pi}_{\boldsymbol{\theta}}} \left[ \mathbf{g}_{D,t,j}^i \right] \right]$$

which is trivially upper-bounded by

$$\mathbb{E}_{s \sim d_{\boldsymbol{\theta}}^t} \left[ \mathbf{Var}_{\mathbf{a} \sim \boldsymbol{\pi}_{\boldsymbol{\theta}}} \left[ \mathbf{g}_{COMA,t,j}^i \right] \right]$$

Now, let us fix a state $s$. We have

$$\mathbf{Var}_{\mathbf{a} \sim \boldsymbol{\pi}_{\boldsymbol{\theta}}} \left[ \mathbf{g}_{COMA,t,j}^i \right] = \mathbf{Var}_{\mathbf{a}^{-i} \sim \boldsymbol{\pi}_{\boldsymbol{\theta}}^{-i}, \mathbf{a}^i \sim \pi_{\boldsymbol{\theta}}^i} \left[ \frac{\partial \log \pi_{\boldsymbol{\theta}}^i(\mathbf{a}^i | s)}{\partial \theta_j^i} A^i(s, \mathbf{a}^{-i}, \mathbf{a}^i) \right]$$

$$\leq \ \mathbb{E}_{\mathbf{a}^{-i} \sim \boldsymbol{\pi}_{\boldsymbol{\theta}}^{-i}, \mathbf{a}^i \sim \pi_{\boldsymbol{\theta}}^i} \left[ \left( \frac{\partial \log \pi_{\boldsymbol{\theta}}^i(\mathbf{a}^i | s)}{\partial \theta_j^i} \right)^2 A^i(s, \mathbf{a}^{-i}, \mathbf{a}^i)^2 \right]$$

$$\leq \ \epsilon_i^2 \, \mathbb{E}_{\mathbf{a}^{-i} \sim \boldsymbol{\pi}_{\boldsymbol{\theta}}^{-i}, \mathbf{a}^i \sim \pi_{\boldsymbol{\theta}}^i} \left[ \left( \frac{\partial \log \pi_{\boldsymbol{\theta}}^i(\mathbf{a}^i | s)}{\partial \theta_j^i} \right)^2 \right] \tag{20}$$

which summing over all components of $\theta^i$ gives

$$\mathbf{Var}_{\mathbf{a} \sim \boldsymbol{\pi}_{\boldsymbol{\theta}}} \left[ \mathbf{g}_{COMA,t}^i \right] \ \leq \ (\epsilon_i B_i)^2$$

Now, applying the reasoning from Equation 19 until the end of the proof of Theorem 1, we arrive at the result

$$\mathbf{Var}_{s_{0:\infty} \sim d_{\boldsymbol{\theta}}^{0:\infty}, \mathbf{a}_{0:\infty} \sim \boldsymbol{\pi}_{\boldsymbol{\theta}}} \left[ \mathbf{g}_{COMA}^i \right] - \mathbf{Var}_{s_{0:\infty} \sim d_{\boldsymbol{\theta}}^{0:\infty}, \mathbf{a}_{0:\infty} \sim \boldsymbol{\pi}_{\boldsymbol{\theta}}} \left[ \mathbf{g}_{D}^i \right] \ \leq \ \frac{(\epsilon_i B_i)^2}{1 - \gamma^2}$$

$\square$

# C Proofs of the Results about Optimal Baselines

In this section of the Appendix we prove the results about optimal baselines, which are those that minimise the CTDE MAPG estimator's variance. We rely on the following variance decomposition

$$\mathbf{Var}_{s_t \sim d_{\boldsymbol{\theta}}^t, \mathbf{a}_t \sim \boldsymbol{\pi}_{\boldsymbol{\theta}}} \left[ \mathbf{g}_{\mathrm{C},t}^i(b) \right] = \mathbf{Var}_{s_t \sim d_{\boldsymbol{\theta}}^t} \left[ \mathbb{E}_{\mathbf{a}_t \sim \boldsymbol{\pi}_{\boldsymbol{\theta}}} \left[ \mathbf{g}_{\mathrm{C},t}^i(b) \right] \right] + \mathbb{E}_{s_t \sim d_{\boldsymbol{\theta}}^t} \left[ \mathbf{Var}_{\mathbf{a}_t \sim \boldsymbol{\pi}_{\boldsymbol{\theta}}} \left[ \mathbf{g}_{\mathrm{C},t}^i(b) \right] \right] \tag{21}$$

$$= \mathbf{Var}_{s_t \sim d_{\boldsymbol{\theta}}^t} \left[ \mathbb{E}_{\mathbf{a}_t \sim \boldsymbol{\pi}_{\boldsymbol{\theta}}} \left[ \mathbf{g}_{\mathrm{C},t}^i(b) \right] \right] + \mathbb{E}_{s_t \sim d_{\boldsymbol{\theta}}^t} \left[ \mathbf{Var}_{\mathbf{a}_t^{-i} \sim \boldsymbol{\pi}_{\boldsymbol{\theta}}^{-i}} \left[ \mathbb{E}_{\mathbf{a}_t^i \sim \boldsymbol{\pi}_{\boldsymbol{\theta}}^i} \left[ \mathbf{g}_{\mathrm{C},t}^i(b) \right] \right] + \mathbb{E}_{\mathbf{a}_t^{-i} \sim \boldsymbol{\pi}_{\boldsymbol{\theta}}^{-i}} \left[ \mathbf{Var}_{\mathbf{a}_t^i \sim \boldsymbol{\pi}_{\boldsymbol{\theta}}^i} \left[ \mathbf{g}_{\mathrm{C},t}^i(b) \right] \right] \right]$$

$$= \underbrace{\mathbf{Var}_{s_t \sim d_{\boldsymbol{\theta}}^t} \left[ \mathbb{E}_{\mathbf{a}_t \sim \boldsymbol{\pi}_{\boldsymbol{\theta}}} \left[ \mathbf{g}_{\mathrm{C},t}^i(b) \right] \right]}_{\text{Variance from state}} + \underbrace{\mathbb{E}_{s_t \sim d_{\boldsymbol{\theta}}^t} \left[ \mathbf{Var}_{\mathbf{a}_t^{-i} \sim \boldsymbol{\pi}_{\boldsymbol{\theta}}^{-i}} \left[ \mathbb{E}_{\mathbf{a}_t^i \sim \boldsymbol{\pi}_{\boldsymbol{\theta}}^i} \left[ \mathbf{g}_{\mathrm{C},t}^i(b) \right] \right] \right]}_{\text{Variance from other agents' actions}} + \underbrace{\mathbb{E}_{s_t \sim d_{\boldsymbol{\theta}}^t, \mathbf{a}_t^{-i} \sim \boldsymbol{\pi}_{\boldsymbol{\theta}}^{-i}} \left[ \mathbf{Var}_{\mathbf{a}_t^i \sim \boldsymbol{\pi}_{\boldsymbol{\theta}}^i} \left[ \mathbf{g}_{\mathrm{C},t}^i(b) \right] \right]}_{\text{Variance from agent } i\text{'s action}}.$$

This decomposition reveals that baselines impact the variance via the local variance $\mathbf{Var}_{\mathbf{a}^i \sim \pi_{\boldsymbol{\theta}}^i} \left[ \mathbf{g}_{\mathrm{C},t}^i(b) \right]$. We rely on this fact in the proofs below.

## C.1 Proof of Theorem 3

**Theorem 3** (Optimal baseline for MAPG). *The optimal baseline (OB) for the MAPG estimator is*

$$b^{optimal}(\mathrm{s}, \mathbf{a}^{-i}) = \frac{\mathbb{E}_{\mathbf{a}^i \sim \pi_{\boldsymbol{\theta}}^i} \left[ \hat{Q}(\mathrm{s}, \mathbf{a}^{-i}, \mathbf{a}^i) \left| \left| \nabla_{\theta^i} \log \pi_{\boldsymbol{\theta}}^i(\mathbf{a}^i|\mathrm{s}) \right| \right|^2 \right]}{\mathbb{E}_{\mathbf{a}^i \sim \pi_{\boldsymbol{\theta}}^i} \left[ \left| \left| \nabla_{\theta^i} \log \pi_{\boldsymbol{\theta}}^i(\mathbf{a}^i|\mathrm{s}) \right| \right|^2 \right]} \tag{7}$$

*Proof.* From the decomposition of the estimator's variance, we know that minimisation of the variance is equivalent to minimisation of the local variance

$$\mathbf{Var}_{\mathbf{a}^i \sim \pi_{\boldsymbol{\theta}}^i} \left[ \left( \hat{Q}(s, \mathbf{a}^{-i}, \mathbf{a}^i) - b \right) \nabla_{\theta^i} \log \pi_{\boldsymbol{\theta}}^i(\mathbf{a}^i|s) \right]$$

For a baseline $b$, we have

$$\mathbf{Var}_{\mathbf{a}^i \sim \pi_{\boldsymbol{\theta}}^i} \left[ \left( \hat{Q}(s, \boldsymbol{a}^{-i}, \mathbf{a}^i) - b) \right) \left( \frac{\partial \log \pi_{\boldsymbol{\theta}}^i(\mathbf{a}^i|s)}{\partial \theta_j^i} \right) \right]$$

$$= \mathbb{E}_{\mathbf{a}^i \sim \pi_{\boldsymbol{\theta}}^i} \left[ \left( \hat{Q}(s, \boldsymbol{a}^{-i}, \mathbf{a}^i) - b) \right)^2 \left( \frac{\partial \log \pi_{\boldsymbol{\theta}}^i(\mathbf{a}^i|s)}{\partial \theta_j^i} \right)^2 \right]$$

$$\qquad - \mathbb{E}_{\mathbf{a}^i \sim \pi_{\boldsymbol{\theta}}^i} \left[ \left( \hat{Q}(s, \boldsymbol{a}^{-i}, \mathbf{a}^i) - b) \right) \left( \frac{\partial \log \pi_{\boldsymbol{\theta}}^i(\mathbf{a}^i|s)}{\partial \theta_j^i} \right) \right]^2$$

$$= \mathbb{E}_{\mathbf{a}^i \sim \pi_{\boldsymbol{\theta}}^i} \left[ \left( \hat{Q}(s, \boldsymbol{a}^{-i}, \mathbf{a}^i) - b) \right)^2 \left( \frac{\partial \log \pi_{\boldsymbol{\theta}}^i(\mathbf{a}^i|s)}{\partial \theta_j^i} \right)^2 \right] \tag{22}$$

$$\qquad - \mathbb{E}_{\mathbf{a}^i \sim \pi_{\boldsymbol{\theta}}^i} \left[ \hat{Q}(s, \boldsymbol{a}^{-i}, \mathbf{a}^i) \left( \frac{\partial \log \pi_{\boldsymbol{\theta}}^i(\mathbf{a}^i|s)}{\partial \theta_j^i} \right) \right]^2$$

as $b$ is a baseline. So in order to minimise variance, we shall minimise the term 22.

$$\mathbb{E}_{\mathbf{a}^i \sim \pi_{\boldsymbol{\theta}}^i} \left[ \left( \hat{Q}(s, \boldsymbol{a}^{-i}, \mathbf{a}^i) - b \right)^2 \left( \frac{\partial \log \pi_{\boldsymbol{\theta}}^i(\mathbf{a}^i|s)}{\partial \theta_j^i} \right)^2 \right]$$

$$= \mathbb{E}_{\mathbf{a}^i \sim \pi_{\boldsymbol{\theta}}^i} \left[ \left( b^2 - 2b\,\hat{Q}(s, \boldsymbol{a}^{-i}, \mathbf{a}^i) + \hat{Q}(s, \boldsymbol{a}^{-i}, \mathbf{a}^i)^2 \right) \left( \frac{\partial \log \pi_{\boldsymbol{\theta}}^i(\mathbf{a}^i|s)}{\partial \theta_j^i} \right)^2 \right]$$

$$= b^2 \, \mathbb{E}_{\mathbf{a}^i \sim \pi_{\boldsymbol{\theta}}^i} \left[ \left( \frac{\partial \log \pi_{\boldsymbol{\theta}}^i(\mathbf{a}^i|s)}{\partial \theta_j^i} \right)^2 \right] - 2b \, \mathbb{E}_{\mathbf{a}^i \sim \pi_{\boldsymbol{\theta}}^i} \left[ \hat{Q}(s, \boldsymbol{a}^{-i}, \mathbf{a}^i) \left( \frac{\partial \log \pi_{\boldsymbol{\theta}}^i(\mathbf{a}^i|s)}{\partial \theta_j^i} \right)^2 \right]$$

$$+ \mathbb{E}_{\mathbf{a}^i \sim \pi_{\boldsymbol{\theta}}^i} \left[ \hat{Q}(s, \boldsymbol{a}^{-i}, \mathbf{a}^i)^2 \left( \frac{\partial \log \pi_{\boldsymbol{\theta}}^i(\mathbf{a}^i|s)}{\partial \theta_j^i} \right)^2 \right]$$

which is a quadratic in $b$. The last term of the quadratic does not depend on $b$, and so it can be treated as a constant. Recalling that the variance of the whole gradient vector $\mathbf{g}^i(b)$ is the sum of variances of its components $\mathbf{g}_j^i(b)$, we obtain it by summing over $j$

$$\mathbf{Var}_{\mathbf{a}^i \sim \pi_{\boldsymbol{\theta}}^i} \left[ \left( \hat{Q}(s, \boldsymbol{a}^{-i}, \mathbf{a}^i) - b \right) \nabla_{\theta^i} \log \pi_{\boldsymbol{\theta}}^i(\mathbf{a}^i|s) \right]$$

$$= \sum_j \left( b^2 \, \mathbb{E}_{\mathbf{a}^i \sim \pi_{\boldsymbol{\theta}}^i} \left[ \left( \frac{\partial \log \pi_{\boldsymbol{\theta}}^i(\mathbf{a}^i|s)}{\partial \theta_j^i} \right)^2 \right] \right.$$

$$\left. - 2b \, \mathbb{E}_{\mathbf{a}^i \sim \pi_{\boldsymbol{\theta}}^i} \left[ \hat{Q}(s, \boldsymbol{a}^{-i}, \mathbf{a}^i) \left( \frac{\partial \log \pi_{\boldsymbol{\theta}}^i(\mathbf{a}^i|s)}{\partial \theta_j^i} \right)^2 \right] + const \right)$$

$$= b^2 \, \mathbb{E}_{\mathbf{a}^i \sim \pi_{\boldsymbol{\theta}}^i} \left[ \left| \left| \nabla_{\theta^i} \log \pi_{\boldsymbol{\theta}}^i(\mathbf{a}^i|s) \right| \right|^2 \right]$$

$$- 2b \, \mathbb{E}_{\mathbf{a}^i \sim \pi_{\boldsymbol{\theta}}^i} \left[ \hat{Q}(s, \boldsymbol{a}^{-i}, \mathbf{a}^i) \left| \left| \nabla_{\theta^i} \log \pi_{\boldsymbol{\theta}}^i(\mathbf{a}^i|s) \right| \right|^2 \right] + const \tag{23}$$

As the leading coefficient is positive, the quadratic achieves the minimum at

$$b^{\text{optimal}} = \frac{\mathbb{E}_{\mathbf{a}^i \sim \pi_{\boldsymbol{\theta}}^i} \left[ \hat{Q}(\text{s}, \boldsymbol{a}^{-i}, \mathbf{a}^i) \left| \left| \nabla_{\theta^i} \log \pi_{\boldsymbol{\theta}}^i(\mathbf{a}^i|s) \right| \right|^2 \right]}{\mathbb{E}_{\mathbf{a}^i \sim \pi_{\boldsymbol{\theta}}^i} \left[ \left| \left| \nabla_{\theta^i} \log \pi_{\boldsymbol{\theta}}^i(\mathbf{a}^i|s) \right| \right|^2 \right]}$$

$$\square$$

### C.2 Remarks about the surrogate optimal baseline

In the paper, we discussed the impracticality of the above baseline. To handle this, we noticed that the policy $\pi_{\boldsymbol{\theta}}^i(\mathbf{a}^i|s)$, at state $s$, is determined by the output layer, $\psi_{\boldsymbol{\theta}}^i(s)$, of an actor neural network. With this representation, in order to handle the impracticality of the above optimal baseline, we considered a minimisation objective, the *surrogate local variance*, given by

$$\mathbf{Var}_{\mathbf{a}^i \sim \pi_{\boldsymbol{\theta}}^i} \left[ \nabla_{\psi_{\boldsymbol{\theta}}^i} \log \pi_{\boldsymbol{\theta}}^i \left( \mathbf{a}^i | \psi_{\boldsymbol{\theta}}^i(s) \right) \left( \hat{Q}(s, \boldsymbol{a}^{-i}, \mathbf{a}^i) - b(s, \boldsymbol{a}^{-i}) \right) \right]$$

As a corollarly to the proof, the surrogate version of the optimal baseline (which we refer to as OB) was proposed, and it is given by

$$b^*(\text{s}, \mathbf{a}^{-i}) = \frac{\mathbb{E}_{\mathbf{a}^i \sim \pi_{\boldsymbol{\theta}}^i} \left[ \hat{Q}(\text{s}, \mathbf{a}^{-i}, \mathbf{a}^i) \left| \left| \nabla_{\psi_{\boldsymbol{\theta}}^i(\text{s})} \log \pi^i \left( \mathbf{a}^i | \psi_{\boldsymbol{\theta}}^i(\text{s}) \right) \right| \right|^2 \right]}{\mathbb{E}_{\mathbf{a}^i \sim \pi_{\boldsymbol{\theta}}^i} \left[ \left| \left| \nabla_{\psi_{\boldsymbol{\theta}}^i(\text{s})} \log \pi^i \left( \mathbf{a} | \psi_{\boldsymbol{\theta}}^i(\text{s}) \right) \right| \right|^2 \right]}.$$

**Remark 6.** *The $x^i_{\psi^i_{\boldsymbol{\theta}}}$ measure, for which $b^*(s, \boldsymbol{a}^{-i}) = \mathbb{E}_{a^i \sim \pi^i_{\boldsymbol{\theta}}} \left[ \hat{Q}(s, \boldsymbol{a}^{-i}, a^i) \right]$, is generally given by*

$$x^i_{\psi^i_{\boldsymbol{\theta}}} \left( \mathrm{a}^i | \mathrm{s} \right) = \frac{\pi^i_{\boldsymbol{\theta}} \left( \mathrm{a}^i | \mathrm{s} \right) \left|\left| \nabla_{\theta^i} \log \pi^i_{\boldsymbol{\theta}} \left( \mathrm{a}^i | \mathrm{s} \right) \right|\right|^2}{\mathbb{E}_{\mathrm{a}^i \sim \pi^i_{\boldsymbol{\theta}}} \left[ \left|\left| \nabla_{\theta^i} \log \pi^i_{\boldsymbol{\theta}} \left( \mathrm{a}^i | \mathrm{s} \right) \right|\right|^2 \right]} \tag{24}$$

Let us introduce the definition of the $\mathrm{softmax}$ function, which is the subject of the next definition. For a vector $\boldsymbol{z} \in \mathbb{R}^d$, we have $\mathrm{softmax}(\boldsymbol{z}) = \left( \frac{e^{z_1}}{\eta}, \ldots, \frac{e^{z_d}}{\eta} \right)$, where $\eta = \sum_{j=1}^d e^{z_j}$. We write $\mathrm{softmax} \left( \psi^i_{\boldsymbol{\theta}}(s) \right) \left( a^i \right) = \frac{\exp\left( \psi^i_{\boldsymbol{\theta}}(s)(a^i) \right)}{\sum_{\tilde{a}^i} \exp\left( \psi^i_{\boldsymbol{\theta}}(s)(\tilde{a}^i) \right)}$.

**Remark 7.** *When the action space is discrete, and the actor's policy is $\pi^i_{\boldsymbol{\theta}} \left( \mathrm{a}^i | \mathrm{s} \right) = \mathrm{softmax} \left( \psi^i_{\boldsymbol{\theta}}(s) \right) \left( a^i \right)$, then the $x^i_{\psi^i_{\boldsymbol{\theta}}}$ measure is given by*

$$x^i_{\psi^i_{\boldsymbol{\theta}}} \left( \mathrm{a}^i | \mathrm{s} \right) = \frac{\pi^i_{\boldsymbol{\theta}} \left( \mathrm{a}^i | \mathrm{s} \right) \left( 1 + \left|\left| \pi^i_{\boldsymbol{\theta}}(s) \right|\right|^2 - 2\pi^i_{\boldsymbol{\theta}} \left( \mathrm{a}^i | \mathrm{s} \right) \right)}{1 - \left|\left| \pi^i_{\boldsymbol{\theta}}(s) \right|\right|^2}$$

*Proof.* As we do not vary states s and parameters $\boldsymbol{\theta}$ in this proof, let us drop them from the notation for $\pi^i_{\boldsymbol{\theta}}$, and $\psi^i_{\boldsymbol{\theta}}(s)$, hence writing $\pi^i \left( \mathrm{a}^i \right) = \mathrm{softmax} \left( \psi^i \right) \left( \mathrm{a}^i \right)$. Let us compute the partial derivatives:

$$\frac{\partial \log \pi^i \left( a^i \right)}{\partial \psi^i \left( \tilde{a}^i \right)} = \frac{\partial \log \mathrm{softmax} \left( \psi^i \right) \left( a^i \right)}{\partial \psi^i \left( \tilde{a}^i \right)} = \frac{\partial}{\partial \psi^i \left( \tilde{a}^i \right)} \left[ \log \frac{\exp \left( \psi^i \left( a^i \right) \right)}{\sum_{\hat{a}^i} \exp \left( \psi^i \left( \hat{a}^i \right) \right)} \right]$$

$$= \frac{\partial}{\partial \psi^i \left( \tilde{a}^i \right)} \left[ \psi^i \left( a^i \right) - \log \sum_{\hat{a}^i} \exp \left( \psi^i \left( \hat{a}^i \right) \right) \right]$$

$$= \mathbf{I} \left( a^i = \tilde{a}^i \right) - \frac{\exp \left( \psi^i \left( \tilde{a}^i \right) \right)}{\sum_{\hat{a}^i} \exp \left( \psi^i \left( \hat{a}^i \right) \right)} = \mathbf{I} \left( a^i = \tilde{a}^i \right) - \pi^i \left( \tilde{a}^i \right)$$

where $\mathbf{I}$ is the indicator function, taking value $1$ if the stetement input to it is true, and $0$ otherwise. Taking $\boldsymbol{e}_k$ to be the standard normal vector with $1$ in $k^{\mathrm{th}}$ entry, we have the gradient

$$\nabla_{\psi^i} \log \pi^i \left( a^i \right) = \boldsymbol{e}_{a^i} - \pi^i \tag{25}$$

which has the squared norm

$$\left|\left| \nabla_{\psi^i} \log \pi^i \left( a^i \right) \right|\right|^2 = \left|\left| \boldsymbol{e}_{a^i} - \pi^i \right|\right|^2 = \left( 1 - \pi^i \left( a^i \right) \right)^2 + \sum_{\tilde{a}^i \neq a^i} \left( -\pi^i \left( \tilde{a}^i \right) \right)^2$$

$$= 1 + \sum_{\tilde{a}^i} \left( -\pi^i \left( \tilde{a}^i \right) \right)^2 - 2\pi^i \left( a^i \right) = 1 + \left|\left| \pi^i \right|\right|^2 - 2\pi^i \left( a^i \right).$$

The expected value of this norm is

$$\mathbb{E}_{\mathrm{a}^i \sim \pi^i} \left[ 1 + \left|\left| \pi^i \right|\right|^2 - 2\pi^i \left( a^i \right) \right] = 1 + \left|\left| \pi^i \right|\right|^2 - \mathbb{E}_{\mathrm{a}^i \sim \pi^i} \left[ 2\pi^i \left( a^i \right) \right]$$

$$= 1 + \left|\left| \pi^i \right|\right|^2 - 2 \sum_{\tilde{a}^i} \left( \pi^i \left( a^i \right) \right)^2 = 1 - \left|\left| \pi^i \right|\right|^2$$

which combined with Equation 24 finishes the proof. $\qquad \square$

## C.3 Proof of Theorem 4

**Theorem 4.** *The excess surrogate local variance for baseline $b$ satisfies*

$$\Delta \mathbf{Var}(b) = \left( b - b^*(s, \boldsymbol{a}^{-i}) \right)^2 \mathbb{E}_{\mathrm{a}^i \sim \pi^i_{\boldsymbol{\theta}}} \left[ \left|\left| \nabla_{\psi^i_{\boldsymbol{\theta}}} \log \pi^i \left( \mathrm{a}^i | \psi^i_{\boldsymbol{\theta}}(s) \right) \right|\right|^2 \right]$$

*In particular, the excess variance of the vanilla MAPG and COMA estimators satisfy*

$$\Delta \mathbf{Var}_{MAPG} \leq D_i^2 \left( \mathbf{Var}_{\mathrm{a}^i \sim \pi^i_{\boldsymbol{\theta}}} \left[ A^i_{\boldsymbol{\theta}}(s, \boldsymbol{a}^{-i}, a^i) \right] + Q^{-i}_{\boldsymbol{\theta}}(s, \boldsymbol{a}^{-i})^2 \right) \leq D_i^2 \left( \epsilon_i^2 + \left[ \frac{\beta}{1 - \gamma} \right]^2 \right)$$

$$\Delta \mathbf{Var}_{COMA} \leq D_i^2 \, \mathbf{Var}_{\mathrm{a}^i \sim \pi^i_{\boldsymbol{\theta}}} \left[ A^i_{\boldsymbol{\theta}}(s, \boldsymbol{a}^{-i}, a^i) \right] \leq (\epsilon_i D_i)^2$$

*where $D_i = \sup_{a^i} \left|\left| \nabla_{\psi^i_{\boldsymbol{\theta}}} \log \pi^i_{\boldsymbol{\theta}} \left( a^i | \psi^i_{\boldsymbol{\theta}}(s) \right) \right|\right|$, and $\epsilon_i = \sup_{s, \boldsymbol{a}^{-i}, a^i} \left| A^i_{\boldsymbol{\theta}}(s, \boldsymbol{a}^{-i}, a^i) \right|$.*

*Proof.* The first part of the theorem (the formula for excess variance) follows from Equation 23. For the rest of the statements, it suffices to show the first of each inequalities, as the later ones follow directly from the fact that $|Q_{\boldsymbol{\theta}}(s, \boldsymbol{a})| \le \frac{\beta}{1-\gamma}$, $\mathbf{Var}_{\mathrm{a}^i \sim \pi_{\boldsymbol{\theta}}^i}\left[A_{\boldsymbol{\theta}}^i(s, \boldsymbol{a}^{-i}, \mathrm{a}^i)\right] = \mathbb{E}_{\mathrm{a}^i \sim \pi_{\boldsymbol{\theta}}^i}\left[A_{\boldsymbol{\theta}}^i(s, \boldsymbol{a}^{-i}, \mathrm{a}^i)^2\right]$, and the definition of $\epsilon_i$. Let us first derive the bounds for $\Delta\mathbf{Var}_{\mathrm{MAPG}}$. Let us, for short-hand, define

$$c_{\boldsymbol{\theta}}^i := \mathbb{E}_{\mathrm{a}^i \sim \pi_{\boldsymbol{\theta}}^i}\left[\left|\left|\nabla_{\psi_{\boldsymbol{\theta}}^i} \log \pi_{\boldsymbol{\theta}}^i\left(\mathrm{a}^i | \psi_{\boldsymbol{\theta}}^i\right)\right|\right|^2\right]$$

We have

$$\Delta\mathbf{Var}_{\mathrm{MAPG}} = \Delta\mathbf{Var}(0) = c_{\boldsymbol{\theta}}^i\, b^*(s, \boldsymbol{a}^{-i})^2 = c_{\boldsymbol{\theta}}^i\, \mathbb{E}_{\mathrm{a}^i \sim x_{\psi_{\boldsymbol{\theta}}^i}^i}\left[\hat{Q}(s, \boldsymbol{a}^{-i}, \mathrm{a}^i)\right]^2$$

$$\le c_{\boldsymbol{\theta}}^i\, \mathbb{E}_{\mathrm{a}^i \sim x_{\psi_{\boldsymbol{\theta}}^i}^i}\left[\hat{Q}(s, \boldsymbol{a}^{-i}, \mathrm{a}^i)^2\right] = c_{\boldsymbol{\theta}}^i \mathbb{E}_{\mathrm{a}^i \sim \pi_{\boldsymbol{\theta}}^i}\left[\frac{\hat{Q}(s, \boldsymbol{a}^{-i}, \mathrm{a}^i)^2 \left|\left|\nabla_{\psi_{\boldsymbol{\theta}}^i} \log \pi_{\boldsymbol{\theta}}^i\left(\mathrm{a}^i | \psi_{\boldsymbol{\theta}}^i\right)\right|\right|^2}{c_{\boldsymbol{\theta}}^i}\right]$$

$$= \mathbb{E}_{\mathrm{a}^i \sim \pi_{\boldsymbol{\theta}}^i}\left[\hat{Q}(s, \boldsymbol{a}^{-i}, \mathrm{a}^i)^2 \left|\left|\nabla_{\psi_{\boldsymbol{\theta}}^i} \log \pi_{\boldsymbol{\theta}}^i\left(\mathrm{a}^i | \psi_{\boldsymbol{\theta}}^i(s)\right)\right|\right|^2\right]$$

$$\le \mathbb{E}_{\mathrm{a}^i \sim \pi_{\boldsymbol{\theta}}^i}\left[\hat{Q}(s, \boldsymbol{a}^{-i}, \mathrm{a}^i)^2 D_i^2\right]$$

$$= D_i^2 \left(\mathbb{E}_{\mathrm{a}^i \sim \pi_{\boldsymbol{\theta}}^i}\left[\hat{Q}(s, \boldsymbol{a}^{-i}, \mathrm{a}^i)^2\right] - \mathbb{E}_{\mathrm{a}^i \sim \pi_{\boldsymbol{\theta}}^i}\left[\hat{Q}(s, \boldsymbol{a}^{-i}, \mathrm{a}^i)\right]^2 + \mathbb{E}_{\mathrm{a}^i \sim \pi_{\boldsymbol{\theta}}^i}\left[\hat{Q}(s, \boldsymbol{a}^{-i}, \mathrm{a}^i)\right]^2\right)$$

$$= D_i^2 \left(\mathbf{Var}_{\mathrm{a}^i \sim \pi_{\boldsymbol{\theta}}^i}\left[\hat{Q}(s, \boldsymbol{a}^{-i}, \mathrm{a}^i)\right] + \hat{Q}^{-i}(s, \boldsymbol{a}^{-i})^2\right)$$

$$= D_i^2 \left(\mathbf{Var}_{\mathrm{a}^i \sim \pi_{\boldsymbol{\theta}}^i}\left[\hat{A}^i(s, \boldsymbol{a}^{-i}, \mathrm{a}^i)\right] + \hat{Q}^{-i}(s, \boldsymbol{a}^{-i})^2\right)$$

which finishes the proof for MAPG. For COMA, we have

$$\Delta\mathbf{Var}_{\mathrm{COMA}} = \Delta\mathbf{Var}\left(\hat{Q}^{-i}(s, \boldsymbol{a}^{-i})\right) = c_{\boldsymbol{\theta}}^i\left(b^*(s, \boldsymbol{a}^{-i}) - \hat{Q}^{-i}(s, \boldsymbol{a}^{-i})\right)^2$$

$$= c_{\boldsymbol{\theta}}^i\left(\mathbb{E}_{\mathrm{a}^i \sim x_{\psi_{\boldsymbol{\theta}}^i}^i}\left[\hat{Q}(s, \boldsymbol{a}^{-i}, \mathrm{a}^i)\right] - \hat{Q}^{-i}(s, \boldsymbol{a}^{-i})\right)^2$$

$$= c_{\boldsymbol{\theta}}^i \mathbb{E}_{\mathrm{a}^i \sim x_{\psi_{\boldsymbol{\theta}}^i}^i}\left[\hat{Q}(s, \boldsymbol{a}^{-i}, \mathrm{a}^i) - \hat{Q}^{-i}(s, \boldsymbol{a}^{-i})\right]^2$$

$$= c_{\boldsymbol{\theta}}^i \mathbb{E}_{\mathrm{a}^i \sim x_{\psi_{\boldsymbol{\theta}}^i}^i}\left[\hat{A}^i(s, \boldsymbol{a}^{-i}, \mathrm{a}^i)\right]^2$$

$$\le c_{\boldsymbol{\theta}}^i \mathbb{E}_{\mathrm{a}^i \sim x_{\psi_{\boldsymbol{\theta}}^i}^i}\left[\hat{A}^i(s, \boldsymbol{a}^{-i}, \mathrm{a}^i)^2\right]$$

$$= c_{\boldsymbol{\theta}}^i \mathbb{E}_{\mathrm{a}^i \sim \pi_{\boldsymbol{\theta}}^i}\left[\hat{A}^i(s, \boldsymbol{a}^{-i}, \mathrm{a}^i)^2 \frac{\left|\left|\nabla_{\psi_{\boldsymbol{\theta}}^i} \log \pi_{\boldsymbol{\theta}}^i\left(\mathrm{a}^i | \psi_{\boldsymbol{\theta}}^i(s)\right)\right|\right|^2}{c_{\boldsymbol{\theta}}^i}\right]$$

$$= \mathbb{E}_{\mathrm{a}^i \sim \pi_{\boldsymbol{\theta}}^i}\left[\hat{A}^i(s, \boldsymbol{a}^{-i}, \mathrm{a}^i)^2 \left|\left|\nabla_{\psi_{\boldsymbol{\theta}}^i} \log \pi_{\boldsymbol{\theta}}^i\left(\mathrm{a}^i | \psi_{\boldsymbol{\theta}}^i(s)\right)\right|\right|^2\right]$$

$$\le \mathbb{E}_{\mathrm{a}^i \sim \pi_{\boldsymbol{\theta}}^i}\left[D_i^2 \hat{A}^i(s, \boldsymbol{a}^{-i}, \mathrm{a}^i)^2\right] \le (\epsilon_i D_i)^2$$

which finishes the proof. $\square$

## D   Pytorch Implementations of the Optimal Baseline

First, we import necessary packages, which are **PyTorch** [20], and its **nn.functional** sub-package. These are standard Deep Learning packages used in RL [1].

```python
import torch, torch.nn.functional as F
```

We then implement a simple method that normalises a row vector, so that its (non-negative) entries sum up to 1, making the vector a probability distribution.

```python
# x: batch of row vectors to normalise to probability mass
normalize = lambda x: F.normalize(x, p=1, dim=-1)
```

The **discrete** OB is an exact dot product between the measure $x^i_{\psi^i_\theta}$, and available values of $\hat{Q}$.

```python
# q: Q values of actions of agent i
# pi: policy of agent i
def optimal_baseline(q, pi):
    M = torch.norm(pi, dim=-1, keepdim = True) ** 2 + 1
    xweight = normalize( (M - 2 * pi) * pi )
    return (xweight * q).sum(-1)
```

In the **continuos** case, the measure $x^i_{\psi^i_\theta}$ and $Q$-values can only be sampled at finitely many points.

```python
# a: sampled actions of agent i
# q: Q values of the sampled actions
# mu, std: parameters of the Gaussian policy of agent i
def optimal_baseline(a, q, mu, std):
    mu_term = torch.norm((a - mu)/std**2, dim=-1)
    std_term = torch.norm(((a - mu)**2 - std**2)/std**3, dim=-1)
    xweight = normalize(mu_term**2 + std_term**2)
    return (xweight * q).sum(-1)
```

We can incorporate it into our MAPG algorith by simply replacing the values of advantage with the values of X, in the buffer. Below, we present a discrete example

```python
# compute the policy and sample an action from it
a, pi = actor(obs)
q = critic(obs)

#compute OB
ob = optimal_baseline(q, pi)

# use OB to construct the loss
q = q.gather(-1, a)
pi = pi.gather(-1, a)
X = q - ob
loss = -(X * torch.log(pi)).mean()
```

and a continuos one

```python
# normal sampling step, where log_pi is the log probability of a
a, log_pi = actor(obs, deterministic=False)
q = critic(obs, a)

# resample m (e.g., m=1000) actions for the observation
obs_m = obs.unsqueeze(0).repeat(m, 1)
a_m, mu_m, std_m = actor(obs, deterministic=False)

# approximate OB
q_m = critic(obs, a_m)
ob = optimal_baseline(a_m, q_m, mu_m, std_m)

# use OB to construct the loss
X = q - ob
loss = -(X * log_pi).mean()
```

## E   Computations for the Numerical Toy Example

Here we prove that the quantities in table are filled properly.

| $a^i$ | $\psi_{\boldsymbol{\theta}}^i(a^i)$ | $\pi_{\boldsymbol{\theta}}^i(a^i)$ | $x_{\psi_{\boldsymbol{\theta}}^i}^i(a^i)$ | $\hat{Q}(\boldsymbol{a}^{-i}, a^i)$ | $\hat{A}^i(\boldsymbol{a}^{-i}, a^i)$ | $\hat{X}^i(\boldsymbol{a}^{-i}, a^i)$ | Method | Variance |
|---|---|---|---|---|---|---|---|---|
| 1 | $\log 8$ | 0.8 | 0.14 | 2 | $-9.7$ | $-41.71$ | MAPG | **1321** |
| 2 | 0 | 0.1 | 0.43 | 1 | $-10.7$ | $-42.71$ | COMA | **1015** |
| 3 | 0 | 0.1 | 0.43 | 100 | 88.3 | 56.29 | OB | **673** |

*Proof.* In this proof, for convenience, the multiplication and exponentiation of vectors is element-wise. Firstly, we trivially obtain the column $\pi_{\boldsymbol{\theta}}^i(a^i)$, by taking $\mathrm{softmax}$ over $\psi_{\boldsymbol{\theta}}^i(a^i)$. This allows us to compute the counterfactual baseline of COMA, which is

$$\hat{Q}^{-i}(\boldsymbol{a}^{-i}) = \mathbb{E}_{a^i \sim \pi_{\boldsymbol{\theta}}^i}\left[\hat{Q}\left(\boldsymbol{a}^{-i}, a^i\right)\right] = \sum_{a^i=1}^{3} \pi_{\boldsymbol{\theta}}^i(a^i)\hat{Q}\left(\boldsymbol{a}^{-i}, a^i\right)$$

$$= 0.8 \times 2 + 0.1 \times 1 + 0.1 \times 100 = 1.6 + 0.1 + 10 = 11.7$$

By subtracting this value from the column $\hat{Q}(\boldsymbol{a}^{-i}, a^i)$, we obtain the column $\hat{A}^i(\boldsymbol{a}^{-i}, a^i)$.

Let us now compute the column of $x_{\psi_{\boldsymbol{\theta}}^i}^i$. For this, we use Remark 7. We have

$$\left|\left|\pi_{\boldsymbol{\theta}}^i\right|\right|^2 = 0.8^2 + 0.1^2 + 0.1^2 = 0.66$$

and $1 + \left|\left|\pi_{\boldsymbol{\theta}}^i\right|\right|^2 - 2\pi_{\boldsymbol{\theta}}^i(a^i) = 1.66 - 2\pi_{\boldsymbol{\theta}}^i(a^i)$, which is 0.06 for $a^i = 1$, and 1.46 when $a^i = 2, 3$. For $a^i = 1$, we have that

$$\pi_{\boldsymbol{\theta}}^i(a^i)\left(1 + \left|\left|\pi_{\boldsymbol{\theta}}^i\right|\right|^2 - 2\pi_{\boldsymbol{\theta}}^i(a^i)\right) = 0.8 \times 0.06 = 0.048$$

and for $a^i = 2, 3$, we have

$$\pi_{\boldsymbol{\theta}}^i(a^i)\left(1 + \left|\left|\pi_{\boldsymbol{\theta}}^i\right|\right|^2 - 2\pi_{\boldsymbol{\theta}}^i(a^i)\right) = 0.1 \times 1.46 = 0.146$$

We obtain the column $x_{\boldsymbol{\theta}}^i(a^i)$ by normalising the vector $(0.048, 0.146, 0.146)$. Now, we can compute OB, which is the dot product of the columns $x_{\psi_{\boldsymbol{\theta}}^i}^i(a^i)$ and $\hat{Q}(\boldsymbol{a}^{-i}, a^i)$

$$b^*(\boldsymbol{a}^{-i}) = 0.14 \times 2 + 0.43 \times 1 + 0.43 \times 100 = 0.28 + 0.43 + 43 = 43.71$$

We obtain the column $\hat{X}^i(\boldsymbol{a}^{-i}, a^i)$ after subtracting $b^*(\boldsymbol{a}^{-i})$ from the column $\hat{Q}(\boldsymbol{a}^{-i}, a^i)$.

Now, we can compute and compare the variances of vanilla MAPG, COMA, and OB. The surrogate local variance of an MAPG estimator $\mathbf{g}^i(b)$ is

$$\mathbf{Var}_{a^i \sim \pi_{\boldsymbol{\theta}}^i}\left[\mathbf{g}^i(b)\right] = \mathbf{Var}_{a^i \sim \pi_{\boldsymbol{\theta}}^i}\left[\left(\hat{Q}^i\left(\boldsymbol{a}^{-i}, a^i\right) - b(\boldsymbol{a}^{-i})\right)\nabla_{\psi_{\boldsymbol{\theta}}^i}\log\pi_{\theta}^i(a^i)\right]$$

$$= \mathrm{sum}\left(\mathbb{E}_{a^i \sim \pi_{\boldsymbol{\theta}}^i}\left[\left(\left[\hat{Q}^i\left(\boldsymbol{a}^{-i}, a^i\right) - b(\boldsymbol{a}^{-i})\right]\nabla_{\psi_{\boldsymbol{\theta}}^i}\log\pi_{\theta}^i(a^i)\right)^2\right] - \mathbb{E}_{a^i \sim \pi_{\boldsymbol{\theta}}^i}\left[\left(\hat{Q}^i\left(\boldsymbol{a}^{-i}, a^i\right) - b(\boldsymbol{a}^{-i})\right)\nabla_{\psi_{\boldsymbol{\theta}}^i}\log\pi_{\theta}^i(a^i)\right]^2\right)$$

$$= \mathrm{sum}\left(\mathbb{E}_{a^i \sim \pi_{\boldsymbol{\theta}}^i}\left[\left(\left[\hat{Q}^i\left(\boldsymbol{a}^{-i}, a^i\right) - b(\boldsymbol{a}^{-i})\right]\nabla_{\psi_{\boldsymbol{\theta}}^i}\log\pi_{\theta}^i(a^i)\right)^2\right]\right) - \mathrm{sum}\left(\mathbb{E}_{a^i \sim \pi_{\boldsymbol{\theta}}^i}\left[\hat{Q}^i\left(\boldsymbol{a}^{-i}, a^i\right)\nabla_{\psi_{\boldsymbol{\theta}}^i}\log\pi_{\theta}^i(a^i)\right]^2\right)$$

where "sum" is taken element-wise. The last equality follows be linearity of element-wise summing, and the fact that $b$ is a baseline. We compute the variance of vanilla MAPG ($\mathbf{g}_{\mathrm{MAPG}}^i$), COMA ($\mathbf{g}_{\mathrm{COMA}}^i$),

and OB ($\mathbf{g}_X^i$). Let us derive the first moment, which is the same for all methods

$$
\mathbb{E}_{\mathrm{a}^i \sim \pi_{\boldsymbol{\theta}}^i} \left[ \hat{Q} \left( \boldsymbol{a}^{-i}, \mathrm{a}^i \right) \nabla_{\psi_{\boldsymbol{\theta}}^i} \log \pi_{\boldsymbol{\theta}}^i(\mathrm{a}^i) \right] = \sum_{a^i=1}^{3} \pi_{\boldsymbol{\theta}}^i(a^i) \hat{Q} \left( \boldsymbol{a}^{-i}, a^i \right) \nabla_{\psi_{\boldsymbol{\theta}}^i} \log \pi_{\theta}^i(a^i)
$$

recalling Equation 25

$$
= \sum_{a^i=1}^{3} \pi_{\boldsymbol{\theta}}^i(a^i) \hat{Q} \left( \boldsymbol{a}^{-i}, a^i \right) \left( \boldsymbol{e}_{a^i} - \pi_{\boldsymbol{\theta}}^i \right)
$$

$$
= 0.8 \times 2 \times \begin{bmatrix} 0.2 \\ -0.1 \\ -0.1 \end{bmatrix} + 0.1 \times 1 \times \begin{bmatrix} -0.8 \\ 0.9 \\ -0.1 \end{bmatrix} + 0.1 \times 100 \times \begin{bmatrix} -0.8 \\ -0.1 \\ 0.9 \end{bmatrix}
$$

$$
= \begin{bmatrix} 0.32 \\ -0.16 \\ -0.16 \end{bmatrix} + \begin{bmatrix} -0.08 \\ 0.09 \\ -0.01 \end{bmatrix} + \begin{bmatrix} -8 \\ -1 \\ 9 \end{bmatrix} = \begin{bmatrix} -7.76 \\ -1.07 \\ 8.83 \end{bmatrix}
$$

Now, let's compute the second moment for each of the methods, starting from vanilla MAPG

$$
\mathbb{E}_{\mathrm{a}^i \sim \pi_{\boldsymbol{\theta}}^i} \left[ \hat{Q} \left( \boldsymbol{a}^{-i}, \mathrm{a}^i \right)^2 \left( \nabla_{\psi_{\boldsymbol{\theta}}^i} \log \pi_{\boldsymbol{\theta}}^i(\mathrm{a}^i) \right)^2 \right]
$$

$$
= \sum_{a^i=1}^{3} \pi_{\boldsymbol{\theta}}^i(a^i) \hat{Q} \left( \boldsymbol{a}^{-i}, a^i \right)^2 \left( \nabla_{\psi_{\boldsymbol{\theta}}^i} \log \pi_{\boldsymbol{\theta}}^i(a^i) \right)^2
$$

$$
= \sum_{a^i=1}^{3} \pi_{\boldsymbol{\theta}}^i(a^i) \hat{Q} \left( \boldsymbol{a}^{-i}, a^i \right)^2 \left( \boldsymbol{e}_{a^i} - \pi_{\boldsymbol{\theta}}^i \right)^2
$$

$$
= 0.8 \times 2^2 \times \begin{bmatrix} 0.2 \\ -0.1 \\ -0.1 \end{bmatrix}^2 + 0.1 \times 1^2 \times \begin{bmatrix} -0.8 \\ 0.9 \\ -0.1 \end{bmatrix}^2 + 0.1 \times 100^2 \times \begin{bmatrix} -0.8 \\ -0.1 \\ 0.9 \end{bmatrix}^2
$$

$$
= 0.8 \times 4 \times \begin{bmatrix} 0.04 \\ 0.01 \\ 0.01 \end{bmatrix} + 0.1 \times \begin{bmatrix} 0.64 \\ 0.81 \\ 0.01 \end{bmatrix} + 0.1 \times 10000 \times \begin{bmatrix} 0.64 \\ 0.01 \\ 0.81 \end{bmatrix}
$$

$$
= \begin{bmatrix} 0.128 \\ 0.032 \\ 0.032 \end{bmatrix} + \begin{bmatrix} 0.064 \\ 0.081 \\ 0.001 \end{bmatrix} + \begin{bmatrix} 640 \\ 10 \\ 810 \end{bmatrix} = \begin{bmatrix} 640.192 \\ 10.113 \\ 810.033 \end{bmatrix}
$$

We have

$$
\mathbf{Var}_{\mathrm{a}^i \sim \pi_{\boldsymbol{\theta}}^i} \left[ \mathbf{g}_{\mathrm{MAPG}}^i \right]
$$

$$
= \mathbb{E}_{\mathrm{a}^i \sim \pi_{\boldsymbol{\theta}}^i} \left[ \hat{Q} \left( \boldsymbol{a}^{-i}, \mathrm{a}^i \right)^2 \left( \nabla_{\psi_{\boldsymbol{\theta}}^i} \log \pi_{\boldsymbol{\theta}}^i(a^i) \right)^2 \right]
$$

$$
- \mathbb{E}_{\mathrm{a}^i \sim \pi_{\boldsymbol{\theta}}^i} \left[ \hat{Q} \left( \boldsymbol{a}^{-i}, \mathrm{a}^i \right) \nabla_{\psi_{\boldsymbol{\theta}}^i} \log \pi_{\boldsymbol{\theta}}^i(a^i) \right]^2
$$

$$
= \begin{bmatrix} 640.192 \\ 10.113 \\ 810.033 \end{bmatrix} - \begin{bmatrix} -7.76 \\ -1.07 \\ 8.83 \end{bmatrix}^2 = \begin{bmatrix} 640.192 \\ 10.113 \\ 810.033 \end{bmatrix} - \begin{bmatrix} 60.2176 \\ 1.1449 \\ 77.9689 \end{bmatrix} = \begin{bmatrix} 579.9744 \\ 8.968 \\ 732.064 \end{bmatrix}
$$

So the variance of vanilla MAPG in this case is

$$
\mathbf{Var}_{\mathrm{a}^i \sim \pi_{\boldsymbol{\theta}}^i} \left[ \mathbf{g}_{\mathrm{MAPG}}^i \right] = 1321.007
$$

Let's now deal with COMA

$$\mathbb{E}_{a^i \sim \pi_{\boldsymbol{\theta}}^i} \left[ \hat{A}^i \left( \boldsymbol{a}^{-i}, a^i \right)^2 \left( \nabla_{\psi_{\boldsymbol{\theta}}^i} \log \pi_{\boldsymbol{\theta}}^i(a^i) \right)^2 \right]$$

$$= \sum_{a^i=1}^{3} \pi_{\boldsymbol{\theta}}^i(a^i) \hat{A}^i \left( \boldsymbol{a}^{-i}, a^i \right)^2 \left( \nabla_{\psi_{\boldsymbol{\theta}}^i} \log \pi_{\boldsymbol{\theta}}^i(a^i) \right)^2$$

$$= \sum_{a^i=1}^{3} \pi_{\boldsymbol{\theta}}^i(a^i) \hat{A}^i \left( \boldsymbol{a}^{-i}, a^i \right)^2 \left( \boldsymbol{e}_{a^i} - \pi_{\boldsymbol{\theta}}^i \right)^2$$

$$= 0.8 \times (-9.7)^2 \times \begin{bmatrix} 0.2 \\ -0.1 \\ -0.1 \end{bmatrix}^2 + 0.1 \times (-10.7)^2 \times \begin{bmatrix} -0.8 \\ 0.9 \\ -0.1 \end{bmatrix}^2 + 0.1 \times 88.3^2 \times \begin{bmatrix} -0.8 \\ -0.1 \\ 0.9 \end{bmatrix}^2$$

$$= 0.8 \times 94.09 \times \begin{bmatrix} 0.04 \\ 0.01 \\ 0.01 \end{bmatrix} + 0.1 \times 114.49 \times \begin{bmatrix} 0.64 \\ 0.81 \\ 0.01 \end{bmatrix} + 0.1 \times 7796.89 \times \begin{bmatrix} 0.64 \\ 0.01 \\ 0.81 \end{bmatrix}$$

$$= \begin{bmatrix} 3.011 \\ 0.753 \\ 0.753 \end{bmatrix} + \begin{bmatrix} 2.327 \\ 9.274 \\ 0.114 \end{bmatrix} + \begin{bmatrix} 499.001 \\ 7.797 \\ 631.548 \end{bmatrix} = \begin{bmatrix} 504.339 \\ 17.824 \\ 632.415 \end{bmatrix}$$

We have

$$\mathbb{E}_{a^i \sim \pi_{\boldsymbol{\theta}}^i} \left[ \mathbf{g}_{\text{COMA}}^i \right]$$

$$= \mathbb{E}_{a^i \sim \pi_{\boldsymbol{\theta}}^i} \left[ \hat{A}^i \left( \boldsymbol{a}^{-i}, a^i \right)^2 \left( \nabla_{\theta^i} \log \pi_{\boldsymbol{\theta}}^i(a^i) \right)^2 \right] - \mathbb{E}_{a^i \sim \pi_{\boldsymbol{\theta}}^i} \left[ \hat{A}^i \left( \boldsymbol{a}^{-i}, a^i \right) \nabla_{\theta^i} \log \pi_{\boldsymbol{\theta}}^i(a^i) \right]^2$$

$$= \begin{bmatrix} 504.339 \\ 17.824 \\ 632.415 \end{bmatrix} - \begin{bmatrix} -7.76 \\ -1.07 \\ 8.83 \end{bmatrix}^2 = \begin{bmatrix} 504.339 \\ 17.824 \\ 632.415 \end{bmatrix} - \begin{bmatrix} 60.2176 \\ 1.1449 \\ 77.9689 \end{bmatrix} = \begin{bmatrix} 444.1214 \\ 16.6791 \\ 554.4461 \end{bmatrix}$$

and we have

$$\mathbf{Var}_{a^i \sim \pi_{\boldsymbol{\theta}}^i} \left[ \mathbf{g}_{\text{COMA}}^i \right] = 1015.2466$$

Lastly, we figure out OB

$$\mathbb{E}_{a^i \sim \pi_{\boldsymbol{\theta}}^i} \left[ \hat{X}^i \left( \boldsymbol{a}^{-i}, a^i \right)^2 \left( \nabla_{\psi_{\boldsymbol{\theta}}^i} \log \pi_{\boldsymbol{\theta}}^i(a^i) \right)^2 \right]$$

$$= \sum_{a^i=1}^{3} \pi_{\boldsymbol{\theta}}^i(a^i) \hat{X}^i \left( \boldsymbol{a}^{-i}, a^i \right)^2 \left( \nabla_{\psi_{\boldsymbol{\theta}}^i} \log \pi_{\boldsymbol{\theta}}^i(a^i) \right)^2$$

$$= \sum_{a^i=1}^{3} \pi_{\boldsymbol{\theta}}^i(a^i) \hat{X}^i \left( \boldsymbol{a}^{-i}, a^i \right)^2 \left( \boldsymbol{e}_{a^i} - \pi_{\boldsymbol{\theta}}^i \right)^2$$

$$= 0.8 \times (-41.71)^2 \times \begin{bmatrix} 0.2 \\ -0.1 \\ -0.1 \end{bmatrix}^2 + 0.1 \times (-42.71)^2 \times \begin{bmatrix} -0.8 \\ 0.9 \\ -0.1 \end{bmatrix}^2 + 0.1 \times 56.29^2 \times \begin{bmatrix} -0.8 \\ -0.1 \\ 0.9 \end{bmatrix}^2$$

$$= 0.8 \times 1739.724 \times \begin{bmatrix} 0.04 \\ 0.01 \\ 0.01 \end{bmatrix} + 0.1 \times 1824.144 \times \begin{bmatrix} 0.64 \\ 0.81 \\ 0.01 \end{bmatrix} + 0.1 \times 3168.564 \times \begin{bmatrix} 0.64 \\ 0.01 \\ 0.81 \end{bmatrix}$$

$$= \begin{bmatrix} 55.6712 \\ 13.92 \\ 13.92 \end{bmatrix} + \begin{bmatrix} 116.7452 \\ 147.756 \\ 1.824 \end{bmatrix} + \begin{bmatrix} 202.788 \\ 3.169 \\ 256.654 \end{bmatrix} = \begin{bmatrix} 375.2044 \\ 164.845 \\ 272.398 \end{bmatrix}$$

We have

$$\mathbb{E}_{\mathrm{a}^i \sim \pi_{\boldsymbol{\theta}}^i} \left[ \mathbf{g}_X^i \right]$$

$$= \mathbb{E}_{\mathrm{a}^i \sim \pi_{\boldsymbol{\theta}}^i} \left[ \hat{X}^i \left( \boldsymbol{a}^{-i}, \mathrm{a}^i \right)^2 \left( \nabla_{\psi_{\boldsymbol{\theta}}^i} \log \pi_{\boldsymbol{\theta}}^i(\mathrm{a}^i) \right)^2 \right] - \mathbb{E}_{\mathrm{a}^i \sim \pi_{\boldsymbol{\theta}}^i} \left[ \hat{X}^i \left( \boldsymbol{a}^{-i}, \mathrm{a}^i \right) \nabla_{\psi_{\boldsymbol{\theta}}^i} \log \pi_{\theta}^i(\mathrm{a}^i) \right]^2$$

$$= \begin{bmatrix} 375.2044 \\ 164.845 \\ 272.398 \end{bmatrix} - \begin{bmatrix} -7.76 \\ -1.07 \\ 8.83 \end{bmatrix}^2 = \begin{bmatrix} 375.2044 \\ 164.845 \\ 272.398 \end{bmatrix} - \begin{bmatrix} 60.2176 \\ 1.1449 \\ 77.9689 \end{bmatrix} = \begin{bmatrix} 314.987 \\ 163.7 \\ 194.429 \end{bmatrix}$$

and we have

$$\mathbf{Var}_{\mathrm{a}^i \sim \pi_{\boldsymbol{\theta}}^i} \left[ \mathbf{g}_X^i \right] = 673.116$$

$\square$

## F  Detailed Hyper-parameter Settings for Experiments

In this section, we include the details of our experiments. Their implementations can be found in the following codebase:

https://github.com/morning9393/
Optimal-Baseline-for-Multi-agent-Policy-Gradients.

In COMA experiments, we use the official implementation in their codebase [7]. The only difference between COMA with and without OB is the baseline introduced, that is, the OB or the counterfactual baseline of COMA [7].

Hyper-parameters used for COMA in the SMAC domain.

| Hyper-parameters | 3m | 8m | 2s3z |
|---|---|---|---|
| actor lr | 5e-3 | 1e-2 | 1e-2 |
| critic lr | 5e-4 | 5e-4 | 5e-4 |
| gamma | 0.99 | 0.99 | 0.99 |
| epsilon start | 0.5 | 0.5 | 0.5 |
| epsilon finish | 0.01 | 0.01 | 0.01 |
| epsilon anneal time | 50000 | 50000 | 50000 |
| batch size | 8 | 8 | 8 |
| buffer size | 8 | 8 | 8 |
| target update interval | 200 | 200 | 200 |
| optimizer | RMSProp | RMSProp | RMSProp |
| optim alpha | 0.99 | 0.99 | 0.99 |
| optim eps | 1e-5 | 1e-5 | 1e-5 |
| grad norm clip | 10 | 10 | 10 |
| actor network | rnn | rnn | rnn |
| rnn hidden dim | 64 | 64 | 64 |
| critic hidden layer | 1 | 1 | 1 |
| critic hiddem dim | 128 | 128 | 128 |
| activation | ReLU | ReLU | ReLU |
| eval episodes | 32 | 32 | 32 |

As for Multi-agent PPO, based on the official implementation [45], the original V-based critic is replaced by Q-based critic for OB calculation. Simultaneously, we have not used V-based tricks like the GAE estimator, when either using OB or state value as baselines, for fair comparisons.

We provide the pseudocode of our implementation of Multi-agent PPO with OB. We highlight the novel components of it (those unpresent, for example, in [45]) in colour.

---

**Algorithm 1** Multi-agent PPO with Q-critic and OB

1: Initialize $\theta$ and $\phi$, the parameters for actor $\pi$ and critic $Q$
2: $episode_{max} \leftarrow step_{max}/batch\_size$
3: **while** $episode \leq episode_{max}$ **do**
4:     Set data buffer $\boldsymbol{D} = \{\}$
5:     Get initial states $s_0$ and observations $\boldsymbol{o_0}$
6:     **for** $t = 0$ **to** $batch\_size$ **do**
7:         **for all** agents $i$ **do**
8:             **if** discrete action space **then**
9:                 $a_t^i, p_{\pi,t}^i \leftarrow \pi(o_t^i; \theta)$ // where $p_{\pi,t}^i$ is the probability distribution of available actions
10:             **else if** continuous action space **then**
11:                 $a_t^i, p_{a,t}^i \leftarrow \pi(o_t^i; \theta)$ // where $p_{a,t}^i$ is the probability density of action $a_t^i$
12:             **end if**
13:             $q_t^i \leftarrow Q(s_t, i, a_t^i; \phi)$
14:         **end for**
15:         $s_{t+1}, o_{t+1}^n, r_t \leftarrow$ execute $\{a_t^1 ... a_t^n\}$
16:         **if** discrete action space **then**
17:             Append $[s_t, \boldsymbol{o_t}, \boldsymbol{a_t}, r_t, s_{t+1}, \boldsymbol{o_{t+1}}, \boldsymbol{q_t}, \boldsymbol{p_{\pi,t}}]$ to $\boldsymbol{D}$
18:         **else if** continuous action space **then**
19:             Append $[s_t, \boldsymbol{o_t}, \boldsymbol{a_t}, r_t, s_{t+1}, \boldsymbol{o_{t+1}}, \boldsymbol{q_t}, \boldsymbol{p_{a,t}}]$ to $\boldsymbol{D}$
20:         **end if**
21:     **end for**
        // from now all agents are processed in parallel in $D$
22:     **if** discrete action space **then**
23:         $\boldsymbol{ob} \leftarrow$ optimal_baseline$(\boldsymbol{q}, \boldsymbol{p_\pi})$ // use data from $D$
24:     **else if** continuous action space **then**
25:         Resample $\boldsymbol{a_{t,1...m}}, \boldsymbol{q_{t,1...m}} \sim \boldsymbol{\mu_t}, \boldsymbol{\sigma_t}$ for each $s_t, \boldsymbol{o_t}$
26:         $\boldsymbol{ob} \leftarrow$ optimal_baseline$(\boldsymbol{a}, \boldsymbol{q}, \boldsymbol{\mu}, \boldsymbol{\sigma})$ // use resampled data
27:     **end if**
28:     $\boldsymbol{X} \leftarrow \boldsymbol{q} - \boldsymbol{ob}$
29:     Loss$(\theta) \leftarrow -$mean$(\boldsymbol{X} \cdot \log \boldsymbol{p_a})$
30:     Update $\theta$ with Adam/RMSProp to minimise Loss$(\theta)$
31: **end while**

The critic parameter $\phi$ is trained with TD-learning [34].

---

Hyper-parameters used for Multi-agent PPO in the SMAC domain.

| Hyper-parameters | 3s vs 5z / 5m vs 6m / 6h vs 8z / 27m vs 30m |
|---|---|
| actor lr | 1e-3 |
| critic lr | 5e-4 |
| gamma | 0.99 |
| batch size | 3200 |
| num mini batch | 1 |
| ppo epoch | 10 |
| ppo clip param | 0.2 |
| entropy coef | 0.01 |
| optimizer | Adam |
| opti eps | 1e-5 |
| max grad norm | 10 |
| actor network | mlp |
| hidden layper | 1 |
| hidden layer dim | 64 |
| activation | ReLU |
| gain | 0.01 |
| eval episodes | 32 |
| use huber loss | True |
| rollout threads | 32 |
| episode length | 100 |

Hyper-parameters used for Multi-agent PPO in the Multi-Agent MuJoCo domain.

| Hyper-parameters | Hopper(3x1) | Swimmer(2x1) | HalfCheetah(6x1) | Walker(2x3) |
|---|---|---|---|---|
| actor lr | 5e-6 | 5e-5 | 5e-6 | 1e-5 |
| critic lr | 5e-3 | 5e-3 | 5e-3 | 5e-3 |
| lr decay | 1 | 1 | 0.99 | 1 |
| episode limit | 1000 | 1000 | 1000 | 1000 |
| std x coef | 1 | 10 | 5 | 5 |
| std y coef | 0.5 | 0.45 | 0.5 | 0.5 |
| ob n actions | 1000 | 1000 | 1000 | 1000 |
| gamma | 0.99 | 0.99 | 0.99 | 0.99 |
| batch size | 4000 | 4000 | 4000 | 4000 |
| num mini batch | 40 | 40 | 40 | 40 |
| ppo epoch | 5 | 5 | 5 | 5 |
| ppo clip param | 0.2 | 0.2 | 0.2 | 0.2 |
| entropy coef | 0.001 | 0.001 | 0.001 | 0.001 |
| optimizer | RMSProp | RMSProp | RMSProp | RMSProp |
| momentum | 0.9 | 0.9 | 0.9 | 0.9 |
| opti eps | 1e-5 | 1e-5 | 1e-5 | 1e-5 |
| max grad norm | 0.5 | 0.5 | 0.5 | 0.5 |
| actor network | mlp | mlp | mlp | mlp |
| hidden layper | 2 | 2 | 2 | 2 |
| hidden layer dim | 32 | 32 | 32 | 32 |
| activation | ReLU | ReLU | ReLU | ReLU |
| gain | 0.01 | 0.01 | 0.01 | 0.01 |
| eval episodes | 10 | 10 | 10 | 10 |
| use huber loss | True | True | True | True |
| rollout threads | 4 | 4 | 4 | 4 |
| episode length | 1000 | 1000 | 1000 | 1000 |

For QMIX and COMIX baseline algorithms, we use implementation from their official codebases and keep the performance consistent with the results reported in their original papers [21, 24]. MADDPG is provided along with COMIX, which is derived from its official implementation as well [15].

Hyper-parameters used for QMIX baseline in the SMAC domain.

| Hyper-parameters | 3s vs 5z / 5m vs 6m / 6h vs 8z / 27m vs 30m |
| :---: | :---: |
| critic lr | 5e-4 |
| gamma | 0.99 |
| epsilon start | 1 |
| epsilon finish | 0.05 |
| epsilon anneal time | 50000 |
| batch size | 32 |
| buffer size | 5000 |
| target update interval | 200 |
| double q | True |
| optimizer | RMSProp |
| optim alpha | 0.99 |
| optim eps | 1e-5 |
| grad norm clip | 10 |
| mixing embed dim | 32 |
| hypernet layers | 2 |
| hypernet embed | 64 |
| critic hidden layer | 1 |
| critic hiddem dim | 128 |
| activation | ReLU |
| eval episodes | 32 |

Hyper-parameters used for COMIX baseline in the Multi-Agent MuJoCo domain.

| Hyper-parameters | Hopper(3x1) / Swimmer(2x1) / HalfCheetah(6x1) / Walker(2x3) |
| :---: | :---: |
| critic lr | 0.001 |
| gamma | 0.99 |
| episode limit | 1000 |
| exploration mode | Gaussian |
| start steps | 10000 |
| act noise | 0.1 |
| batch size | 100 |
| buffer size | 1e6 |
| soft target update | True |
| target update tau | 0.001 |
| optimizer | Adam |
| optim eps | 0.01 |
| grad norm clip | 0.5 |
| mixing embed dim | 64 |
| hypernet layers | 2 |
| hypernet embed | 64 |
| critic hidden layer | 2 |
| critic hiddem dim | [400, 300] |
| activation | ReLU |
| eval episodes | 10 |

Hyper-parameters used for MADDPG baseline in the Multi-Agent MuJoCo domain.

| Hyper-parameters | Hopper(3x1) / Swimmer(2x1) / HalfCheetah(6x1) / Walker(2x3) |
|---|---|
| actor lr | 0.001 |
| critic lr | 0.001 |
| gamma | 0.99 |
| episode limit | 1000 |
| exploration mode | Gaussian |
| start steps | 10000 |
| act noise | 0.1 |
| batch size | 100 |
| buffer size | 1e6 |
| soft target update | True |
| target update tau | 0.001 |
| optimizer | Adam |
| optim eps | 0.01 |
| grad norm clip | 0.5 |
| mixing embed dim | 64 |
| hypernet layers | 2 |
| hypernet embed | 64 |
| actor network | mlp |
| hidden layer | 2 |
| hiddem dim | [400, 300] |
| activation | ReLU |
| eval episodes | 10 |