# OpenReview forum: "Settling the Variance of Multi-Agent Policy Gradients"
_NeurIPS.cc/2021/Conference — NeurIPS 2021 Poster_

### Official Review · Reviewer_23LJ · 2021-07-03

**Rating:** 7
**Confidence:** 4

**Summary:**

This paper analyzes the variance of gradient estimates in policy-gradient-based multi-agent RL, specifically for standard multi-agent policy gradient (including both centralized and decentralized training) and for the COMA gradient. The paper then derives an optimal baseline (OB) as well as a tractable surrogate for variance reduction. The non-optimality of the variance of standard MAPG and COMA is shown. Numerically, lower variance of the OB is shown in an analytical example, and performance (in terms of reward and win rate) improvement is shown when the OB is implemented on top of MAPPO and COMA in the discrete-action StarCraft micromanagement scenarios and continuous-action multi-agent MuJoCo.

**Limitations And Societal Impact:**

The paper does mention that deterministic methods and value-function based methods are not in scope.

**Main Review:**

This is a timely paper, given the increasing attention on multi-agent learning and the existence of a number of empirical methods-centric work that builds on policy-gradient methods for MARL. The style of this work should remind a reader of the single-agent counterpart---Variance Reduction for Policy Gradient with Action-Dependent Factorized Baselines, Wu et al, ICLR 2018 (not cited)---but the analysis and multi-agent context are different enough from the single-agent case that this paper has sufficient originality.

Quality meets the bar (but see the detailed questions below). Overall, this work is a good and rare example of a paper in multi-agent RL that derives an algorithmic improvement from theoretical analysis and also provides empirical validation.

One point that the authors should clarify: could the left-hand side of the inequaltiy in Theorem 1 be negative, so that the variance of the decentralized estimator is higher than the centralized one? If not, the paper can be improved if the authors include a conceptual explanation for the reason. The centralized one uses the joint $Q(s,\mathbf{a})$ that receives the joint action as input, while the decentralized one uses $Q^i(s,a^i)$ that itself requires an expectation over the other agents' actions $a^{-i}$. Is there any intuition for one to have higher variance than the other?

The clarity of the paper's notation is appreciated. Providing reductions to known cases also helps a reader to check correctness.

It's not clear what the paper means by line 162: "a MARL problem can be considered as a sum of $n$ RL problems". At best, it's an unnecessary description of Lemma 1. At worst, it could confuse someone into thinking that decentralized training is sufficient for cooperation in MARL, which is obviously false.

It appears that the surrogate optimal baseline in equation (9) is not the one the paper actually uses in practice. Instead, the paper uses the form given in lines 249-250, which is consistent with the description in lines 274-275 that says "firstly, it evaluates the probability measure $x^i_{\psi^i_{\theta}}$, and then computes the expectation ...". It's not very clear how the expression in lines 249-250 is equivalent to equation (9).

In lines 252-253, it's not clear where the "importance weight" arises; perhaps the authors do not mean it in the sense of importance sampling.

The claim in line 314 that OB improves MAPPO on all four tasks is incorrect: there is no significant difference between MAPPO with OB and without OB in 3-agent Hopper, Figure 2a.


Minor points:
1. There are numerous typos that the authors should fix: line 37 "single-agnet", line 236 "this this", line 245 "Hereafater", Table 1 caption "a numerical toy exmaple tha shows", line 272 "This gives as an estimator", line 279 "continuosu"

---------------------------------
Post rebuttal: The authors have responded in detail and addressed my questions, and I am keeping the score for this good paper.

**Time Spent Reviewing:**

5 hours

---

> ### Author Response · Authors · 2021-08-09
> **Response to Reviewer 23LJ**
>
> ### We thank Reviewer 23LJ for his/her 5-hour efforts in offering  constructive comments that will surely turn our paper into a better shape.
>
> * > **Reviewer**:  Overall, this work is a good and rare example of a paper in multi-agent RL that derives an algorithmic improvement from theoretical analysis and also provides empirical validation.
> * **Response**: We greatly appreciate that the reviewer has recognised it. Indeed, our intention is to study and develop our understanding of the MARL problem, and solve it in a way that provably takes advantage of its discovered properties.
>
> 1. > **Reviewer**: A related work “Variance Reduction for Policy Gradient with Action-Dependent Factorized Baselines” [Wu et. al., 2018] was not cited.
>
> * **Response**: We admit that the analysis of optimal action-dependent baseline in [Wu et. al., 2018], though in single-agent cases, is related to our analysis of OB.  We will surely acknowledge this work and cite it in the final version. Moreover, we want to emphasize our novelty in the sense that we study  cooperative MARL problems under both CTDE and DT paradigms; this has not been covered by any no prior works. The differences are dependent on structures that only reveal in MARL: the number of agents, and the local advantage. The results also enabled us to study the properties of baseline tricks in the CTDE framework. As a result, our OB is introduced as a general tool which can be applied to existing MAPG algorithms (e.g. COMA, MAPPO).
>
> 2. > **Reviewer**: The reviewer is interested if the left-hand side of Theorem 1 be negative? Can the variance of the decentralised gradient be higher than that of the centralised one?
>
> * **Response**: The short answer is no. This has been proved in  [Lyu et. al. 2021] (Theorem 2), as we also mentioned in the line 189. In fact, this result also comes up as a corollary of the proof of Theorem 1, mentioned in line 183, where the variance difference is shown to be an expectation over a product of squares of scalars.
>
>     Moreover, we are grateful for your question regarding the underlying intuition; the answer to which is at the essence of the difference between CTDE and DT, and of the credit assignment problem. As mentioned by the reviewer, CTDE and DT gradients differ only by using $Q(s, a^i, a^{-i})$ and $Q^i(s, a^i)$, respectively. The latter is the expectation of the former, $Q^i(s, a^i)=E_{a^i\sim\pi^i_\theta}[ Q(s, a^i, a^{-i})]$, which specifically means that DT depends only on random variables $s$ and $a^i$, while the CTDE depends on extra random variables aggregated into $a^{-i}$. Thus, CTDE admits a "base" variance from $s$ and $a^i$ that DT possesses, plus an extra variance coming from random $a^{-i}$. We are planning to include this intuition in the paper by modifying the sentence from lines 179-182 to the following way:
>
>     (line 179-182) "The goal of our proof strategy was to collate terms $\hat{Q}(s, a^{-i}, a^i)$ and $\hat{Q}^i(s, a^i)$, primarily because the former depends on more random variables, $a^{-i}$, than the latter, making the excess of its variance clear. Moreover, their difference takes a form of a multi-agent advantage, which we extensively studied in previous lemmas."
>
> 3. > **Reviewer**: Saying that a MARL problem is a sum of n RL problems (line 162) may confuse a reader to think that distributed learning is sufficient for cooperation.
>
> * **Response**: With your suggestion in mind, we will amend the paper as follows:
>
>     (line 160) “The lemma reveals that agents contribute to the total return in a sequentially additive way. This intuition leads to an idea of decomposing the variance (...)”.
>
> 4. > **Reviewer**: It is not clear how the baseline from Equation (9) is related to the one used in practice, given in 250, and incorporating measure .
>
> * **Response**: The two baselines are the same. We agree that this might not be immediately clear: in general (not only softmax case) the measure over which the baseline is an expectation (line 250) is
>
>     $x^i_{\psi^i_\theta} (a^i|s) = \frac{ \pi^i_\theta(a^i|s)||\nabla_{\psi^i_\theta}\log\pi^i(a^i|\psi^i_\theta(s))||^2 }{ E_{a^i\sim\pi^i}\left[ ||\nabla_{\psi^i_\theta}\log\pi^i(a^i|\psi^i_\theta(s))||^2 \right]}$.
>
>     Equation (10) under the line 251 essentially describes it for a  practical case of the categorical distribution with  softmax activation in the last layer.
>
> 5. > **Reviewer**: The reviewer is confused what does “importance weight” in line 253 mean?
>
> * **Response**: We are sorry for the confusion. What we want to say is that actions that $\pi^i_\theta$ puts less probability on (thus viewing them as less “important”) can attain more probability in $x^i_{\psi^{i}_\theta}$. We will remove the word “importance” to avoid misunderstanding.
>
> 6. > **Reviewer**: In line 314 authors claim that OB improves the performance of MAPPO on all four MuJoCo tasks, while their performance seems equal on 3-agent Hopper.
>
> * **Response**: We admit that, in terms of total rewards, the two versions of MAPPO perform similarly on the 3-agent Hopper task. We see the improvments in the sense that OB does decrease the variance of  gradient’s norm on every single task (as shown in Table 2), which helps stablise the training.
>
>
> ***Reference***:
>
> - [Wu et. al., 2018] Variance Reduction for Policy Gradient with Action-Dependent Factorized Baselines. ArXiv, abs/1803.07246.
> - [Lyu et. al. 2021] Xueguang Lyu, Yuchen Xiao, Brett Daley, and Christopher Amato. Contrasting centralized and decentralized critics in multi-agent reinforcement learning. In Proceedings of the 20th International Conference on Autonomous Agents and MultiAgent Systems, pages 844–852, 2021.

---

> > ### Comment · Reviewer_23LJ · 2021-08-17
> > **Authors have addressed most of this reviewer's questions**
> >
> > I appreciate the author's detailed replies and the author's proposed edits to the text. I have only two remaining suggestions:
> > 1. If space permits, I suggest that the general form of the measure (over which the optimal baseline is an expectation), which is currently given in Remark 6 in the appendix, should be stated in the main text, so that it is clear that the form on line 250 immediately follows from equation (9).
> > 2. It's commonly assumed that "performance" means a score like "win rate" or "episodic reward" rather than variance, so it would be better if lines 313-314 were edited to be more precise.

---

> > > ### Author Response · Authors · 2021-08-18
> > > **Appreciating further suggestions**
> > >
> > > Dear Reviewer 23LJ
> > >
> > > Likewise, we appreciate the quick response, as well as the detailed instructions for improvement. We are actively working on incorporating the valuable feedback of the reviewers into our paper.
> > >
> > > As for the above two suggestions, suggestion 1 is already addressed in the current version of the final paper. Suggestion 2 has not been incorporated yet, but we see its importance, and so the final version of the paper will address it.
> > >
> > > Many thanks,
> > > Authors

---

### Official Review · Reviewer_uNWS · 2021-07-14

**Rating:** 7
**Confidence:** 3

**Summary:**

This paper theoretically studies the variance of multi-agent policy gradients.

It shows that the extra variance that CTDE MAPG methods have compared to DT methods scales linearly on the number of agents, and quadratically on the local advantage of each agent.

The paper further proposes the optimal baseline technique to reduce the variance of the MAPG estimator. While the analytic form of the optimal baseline is computational challenging to compute, the assumption that the policies are parameterized by neural networks with softmax layer is used to find a surrogate minimisation objective that is more tractable to compute/approximate.

Experiments are conducted to confirm the effectiveness of the optimal baseline technique.

**Limitations And Societal Impact:**

My main concern is that the experimental results are a bit weak in terms of supporting the effectiveness of the optimal baseline technique. It seems to be clear that using the OB can further reduce variance of policy gradients. However, that doesn't seem to be directly translated into an increase of performance, as shown by experiments in SMAC except 2(e).

**Main Review:**

Originality: The results presented in this paper are novel as far as I know.

Quality: The theorems and proofs look correct.

Clarity: The submission was very clearly written and well orgnaized.

Significance: I think this work takes an important step towards understanding policy gradient methods for MARL. The theorem 1 and 2 are very insightful.



**Time Spent Reviewing:**

6

---

> ### Author Response · Authors · 2021-08-09
> **Response to Reviewer uNWS**
>
> ### We thank Reviewer uNWS for his/her 6-hour efforts in offering  constructive comments that will surely turn our paper into a better shape.
>
> 1. > **Reviewer**: Optimal baseline does not always translate into an increase of performance, as shown by experiments in SMAC except 2(e).
> * **Response**: We agree that the current presentation of Figure 2 does not clearly demonstrate the performance gap achieved by OB, this is also possibly because we did not perform any curve smoothing. To address this issue, we have run more random seeds and report the results that average over a sliding window of size $n=10$ (see the anonymous [link](https://drive.google.com/file/d/16hEc2vAO3dXic3Kv5TBAfbV2UjNy-O6o/view?usp=sharing)). Moreover, we remind that SMAC agents do not maximise the average winning rate, but rather the reward emitted by the environment. OB is a technique that is supposed to improve gradient-based reward maximisation. In this  anonymous [link](https://drive.google.com/file/d/1JvVLFTywe_BvtJoWXhOulKI6RmvHjX4U/view?usp=sharing), we provide the learning curves in terms of the average episodic reward, on which the performance gap is larger, which confirms our theory.

---

> > ### Comment · Reviewer_uNWS · 2021-08-22
> > **Thanks for the additional results.**
> >
> > I appreciate the new results provided and will keep my score.

---

> > > ### Author Response · Authors · 2021-08-23
> > > **Thank you for your comments**
> > >
> > > Dear Reviewer uNWS
> > >
> > > Thank you for your suggestions that will improve the quality of our paper, as well as we appreciate your final decision.
> > >
> > > Best regards
> > > Authors

---

### Official Review · Reviewer_uERi · 2021-07-16

**Rating:** 7
**Confidence:** 4

**Summary:**

This paper carefully analyzes the variance of multi-agent policy gradients and derives the multi-agent baseline estimation with the smallest possible variance, which also indeed leads to a lower variance of policy gradients of the state-of-the-art multi-agent PPO algorithm.

**Limitations And Societal Impact:**

Here are a few minor comments.

(1) I couldn't see the confidence interval in Fig2 (e)-(h). I would suggest the authors complete the results in the final version.

(2) I do like the variance analysis. However, these insights only hold under the ideal unbiased case while in practice, the biggest benefit of PPO is due to GAE. The paper could be even stronger if the authors could investigate how to incorporate GAE into the framework.

**Main Review:**

This paper is clearly written with very intuitive explanations and thorough step-by-step derivations on the multi-agent variance. This is, to the best of my knowledge, the first paper that rigorously studies the variance of multi-agent policy gradient methods (although some empirical studies have been indeed investigated). The theoretical analysis starts from the simplest formulation and carefully goes through all the details, which makes the paper particularly easy to follow.

I enjoy reading this paper a lot and this paper indeed tackles an important problem in MARL. I'm happy to see it get accepted.


**Time Spent Reviewing:**

2

---

> ### Author Response · Authors · 2021-08-09
> **Response to Reviewer uERi**
>
> ### We thank Reviewer uERi for his/her 2-hour efforts in offering  constructive comments that will surely turn our paper into a better shape.
>
> 1. > **Reviewer**: Lack of confidence interval in Fig2 (e)-(h).
>
> * **Response**: We acknowledge the shortcoming of this part of our work. We have not managed to run sufficiently many seeds to provide the shadow corresponging to the standard deviation of the win rate.  However, the performance of QMIX that we report, in terms of the median value, is consistent with Figure 14 of [Yu et. al., 2021].  We elaborated the missing work by running more seeds and adjusting our plotting techniques, and provide improved plots under the anonymised [link](https://drive.google.com/file/d/16hEc2vAO3dXic3Kv5TBAfbV2UjNy-O6o/view?usp=sharing).
>
> 2. > **Reviewer**: It would be interesting to see how MAPPO with both OB and GAE works.
>
> * **Response**: We are grateful for the exciting suggestion.  The goal of our experiments was to show the advantage of using OB, we found using a Q-critic and subtracting OB from Q-critic is probably the most straightforwad way. However, to exhaust its practicality, we can incorporate OB into GAE. A naive modification that we came up with is
>
>     $\quad \quad \quad \quad \quad \quad \quad \quad \quad \quad \quad \quad \hat{X}^i_t = \hat{A}_t + \hat{V}(s_t) - ob^i_t$,
>
>     where $\hat{A}_t$ is a GAE, and $ob^i_t$ is OB of agent $i$. We then proceed by maximing the modified PPO-clip objective from line 273. However, we would like to highlight that adoping OB into GAE triggers additional computational complexity. Namely, to implement the simple equation above, in principle, we require both V and Q critics, while the vanilla GAE and the vanilla OB both need only V and Q critic, respectively.
>
>     Nevertheless, we have implemented the idea of GAE+OB and present its performance on one super-hard SMAC map for proof of concept (see the anonymised [link](https://drive.google.com/file/d/1aj1xjEzb_7xXhR0Xcv1AGzXZEk-A6s62/view?usp=sharing) ). Based on the promising result, we will conduct an extensive investigation in future work.
>
> ***Reference***:
>
> - [Yu et. al., 2021] Chao Yu, A. Velu, Eugene Vinitsky, Yu Wang, A. Bayen, and Yi Wu. The surprising effectiveness of mappo in cooperative, multi-agent games. ArXiv, abs/2103.01955, 2021.

---

> > ### Comment · Reviewer_uERi · 2021-08-14
> > **thanks for the additional results**
> >
> > It's exciting to see some preliminary results on GAE with OB. It would be great if the authors could include more experiments in the final draft as suggested in the rebuttal period.
> >
> > I'm also looking forward to seeing how OB performs in a wider range of applications.

---

> > > ### Author Response · Authors · 2021-08-14
> > > **Response to Reviewer uERI**
> > >
> > > Dear Reviewer uERI:
> > >
> > > Thanks for the quick response. And we appreciate the constructive comments you have shared.
> > >
> > > Authors.

---

### Official Review · Reviewer_n8hf · 2021-07-17

**Rating:** 4
**Confidence:** 4

**Summary:**

The paper formally analyzes the variance of multi-agent policy gradients and propose a baseline (and a practical implementation) to reduce the variance. The problem considered is interesting, but the reviewer is concerned about some assumptions used by the authors and is confused about the experimental results.

**Limitations And Societal Impact:**

The authors didn't discuss the limitations and potential impact of the proposed method.

**Main Review:**

### Quality
(1) Assumption: the authors assume that (1) critics (both centralized and decentralized versions) are accurate and can give exact value estimations, and (2) agents can observe the global state $s$. Based on the second assumption, the joint policy is decomposed as a multiplication of local policies. These two assumptions make the theoretical results provided in this paper less surprising.

For example, lemma 1-3 can be obtained by using preliminary math given the second assumption. When Q is accurate, the proof of Theorem 1 does not give much insight, which can be finished by introducing a notation for the upper bound for the gradients and advantages.

(2) In Fig. 2, the variance of the proposed method is very large, especially for complex tasks like SMAC and 2-Agent Walker. It is quite strange because the authors claim that they use an *optimal* baseline for MAPG methods. In contrast, state-of-the-art MAPG methods, like stochastic DOP [Wang et al., ICLR 2021], have much smaller variance.

Moreover, the reviewer is confused about the experimental results. MAPPO performs well on 3s_vs_5z, but the authors report a win rate of 0. And, the confidence interval (or variance or std, the authors do not explain what's this in the main text) in Fig. 2 is quite strange. Why does QMIX not have shaded areas? And, for MAPPO w/o OB, why the intervals are very small when the mean value oscillates significantly. This does not make sense unless only two or three seeds are used when drawing the plots.

### Clarity
(1) Line 43. Not all CTDE PG methods take actions as input to the critic, like COMA. And, the output is the expected return, not the reward.
(2) Line 142, although MADDPG learns deterministic policies, it is a multi-agent policy gradient method. The statement of "These methods are not of our interest, as they learn either a deterministic policy or a value function, thus not in the scope of MAPG methods." is not accurate.

**Time Spent Reviewing:**

5

---

> ### Author Response · Authors · 2021-08-09
> **Response to Reviewer n8hf**
>
> ### We thank Reviewer n8hf for his/her 5-hour efforts in offering  constructive comments that will surely turn our paper into a better shape.
>
> 1. > **Reviewer**: In the variance analysis, the author considers the settings of:  (1) the critics are accurate and give exact  value estimations, and (2) agents observe the global state, and these assumptions make the theoretical results (e.g. Lemma 1-3) less surprising.
>
> * **Response**: In general, the policy gradient process consists of  two components: fitting the critics and updating the policies based on the critics. In this paper, we focus particularly on the second component (thus making Assumption (1)) and theoretically analyse the variance of multi-agent policy gradient estimation. Even under this setting, to our best knowledge, there are no prior works that try to settle the variance of multi-agent policy gradient estimations. One probable reason for that could be a lack of mathematical lever. In our paper, before we arrive at Theorem 1 & 2, we have to introduce novel notions---multi-agent state-action value function
>
>     $Q^{i_1, \dots, i_m}\left(s, a^{(i_1, \dots, i_m)} \right) =
> E_{a^{-(i_1, \dots, i_m)}\sim\pi_\theta^{-(i_1, \dots, i_m)}}\left[ Q\left(s, a^{(i_1, \dots, i_m)}, a^{-(i_1, \dots, i_m)}\right) \right]$
>
>     and multi-agent advantage function
>
>     $A^{j_1, \dots, j_k}\left(s, a^{(i_1, \dots, i_m)}, a^{(j_1, \dots, j_k)}\right) =
> Q^{i_1, \dots, i_m, j_1, \dots, j_k}\left(s, a^{(i_1, \dots, i_m, j_1, \dots, j_k)}\right) - Q^{i_1, \dots, i_m}\left(s, a^{(i_1, \dots, i_m)}\right).$
>
>     The functions describe the contributions to the return that particular subsets of agents have. We believe that these new notions can enable insightful discoveries in cooperative MARL in general for future work. Leveraging on these notions, we propose Lemmas 1-3 and develop their proofs. Therefore, we believe the variance analysis results of Theorem 1 & 2 are non-trivial;  introducing multi-agent state-action value and advantage functions is one of the novelties in this paper.
>
>   $\newline$
>
>     As for Assumption (2), although the notation $\pi^i(a^i|s)$ may suggest a reader full observability, our results do not require it. All we assume is that there is no randomness in what agent $i$ observes given a state $s$. This enables us to say that a mapping from state $s$ to a probability distribution over actions $\pi^i(\cdot|s)$ is well-defined. Such an assumption is valid when the state emits a signal $o^i(s)$ for agent $i$ which, in general, can be referred to as an observation. Indeed, one could write $\pi^i(a^i|o^i(s))$. Such a deterministic observation model was adopted in many literatures, for example, in COMA [Foerster et. al., 2018]. All of our results remain valid in this general partially-observable setting.
>
> 2. > **Reviewer**: The authors claim to use an optimal baseline, while another MAPG method---DOP [Wang et. al, 2021]---achieves lower variance. Is that not a contradiction?
>
> * **Response**: We believe there is no contradiction between our OB with DOP. Firstly, we would like to highlight that the DOP method studies a specific setting in which the Q-function has to follow the decomposition rule as:$Q^{tot}({\tau}, {a}) = \sum_{i}k_i({\tau})Q_i({\tau}_i, a_i) + b({\tau})$. Once trained, the critic  can be decomposed into $Q_i({\tau}_i, a_i)$, so that each agent can use its own action-value $Q_i({\tau}_i, a_i)$ while estimating MAPG. As $Q_i({\tau}_i, a_i)$  does not depend on other agents actions $a^{-i}$, which are random variables, its gradient estimate can have lower variance than the estimator with the extra $a^{-i}$ used as input.
>
>     $\newline$
>
>     Unfortunately, the above Q-function decomposition is not guaranteed to always hold in principle. For instance, the decomposition is invalid when $Q^{tot}$ is multiplicative, rather than linear, in $Q_i$. An example would be a simple one-step game with two agents: the agents take actions $a_1$ and $a_2$ drawn from Gaussian distributions with means $\mu_1$ and $\mu_2$, both with variance $1$. They receive reward $r(a_1, a_2) = a_1 a_2$. In this case, the total Q-function is given as $Q^{tot}(a_1, a_2) = a_1 a_2$. The Q-functions of the two agents are $Q_1(a_1)= a_1\mu_2$ and $Q_2(a_2) = \mu_1 a_2$. In the case with one step the functions $k_i({\tau}_i)$ and $b({\tau})$ are just constants, $k_i$ and $b$. Hence, the right-hand side of the decomposition of DOP is given by $k_1 \mu_2 a_1 + k_2 \mu_1 a_2 + b$, and this does not satisfy the given total Q-function of $Q^{tot}(a_1, a_2)= a_1 a_2$.
>
>     $\newline$
>
>     Thus, to successfully fit $Q^{tot}$, DOP must learn functions $Q_i$ which are not necessarily the true action-values. As a result, the gradient that DOP estimates is not (in principle) equal to the true MAPG. Therefore, DOP can not be considered to have achieved an MAPG estimator with the lowest variance, contradicting the optimality of our OB. Moreover, even if a method was found that learns the true $Q_i$ and achieves lower variance than OB, this would not negate the optimality of our OB; this is because OB minimises the variance of MAPG estimator with the joint action critic, and an estimator that marginalises over $a^{-i}$ should be able to achieve lower variance (i.e., we proved the case for $g_C$ and $g_D$ in Theorem 1). For the above reasons, we see no contradiction.
>
> 3. > **Reviewer**: MAPPO is known to work well on 3s_vs_5z, but the authors report a win rate of 0.
>
> * **Response**: We want to clarify that the MAPPO [Yu et. al., 2021] result we report is the case where MAPPO uses a Q-critic in MAPG estimation. To fairly compare the effectiveness of the proposed OB, we implement the Q-critic-based version of MAPPO to incorporate the OB and then examine the performance improvement it brings (compared to Q-critic without OB). Throughout the comparison, we maintain the same set of hyperparameters  for both MAPPO versions (with and without OB), which we refer to Section F in the supplementary material. Finally, we want to highlight that although the winning rate of MAPPO w/o OB is 0, the episodic reward is noticeably increasing, as we demonstrate on plots under an anonymised [link](https://drive.google.com/file/d/1JvVLFTywe_BvtJoWXhOulKI6RmvHjX4U/view?usp=sharing). Therefore, we believe our implementation of MAPPO is correct.
>
> 4. > **Reviewer**: Lack of shadow (and description of what it stands for) for the baseline method of QMIX in Figure 2.
>
> * **Response**: In all plots, the shadow describes the standard deviation of a performance metric at a given iteration, across different seeds. Due to time constraints, we ran our experiments on these tasks with only a limited number of seeds, which were too few to provide a shadow. However, the performance of QMIX  that we report, in terms of the median value, is consistent with Figure 14 of [Yu et. al., 2021]. To address this issue, we here provide the updated plot with 5 random seeds under the following anonymised [link](https://drive.google.com/file/d/16hEc2vAO3dXic3Kv5TBAfbV2UjNy-O6o/view?usp=sharing).
>
> 5. > **Reviewer**: Inaccurate claim in line 43 about CTDE and line 142 about MADDPG.
>
> * **Response**: Thank you for pointing this out. We will change it as follows:
>
>    -  at line 42. “In CTDE, each agent during training maintains a centralised critic which takes joint information, for example COMA takes in joint state-action pair as input and outputs an estimate of the expected return (...).”
>    - line 141. “These methods are not of our interest, as they learn either a deterministic policy or a value function, thus not in the scope of MAPG methods that learn a stochastic policy.”
>
>
> ***Reference***:
>
> - [Wang et, al., ICLR 2021] Yihan Wang, Beining Han, Tonghan Wang, Heng Dong, and Chongjie Zhang. Off-policy multi-agent decomposed policy gradients. arXiv preprint arXiv:2007.12322, 2020.
> - [Yu et, al. 2021] Chao Yu, A. Velu, Eugene Vinitsky, Yu Wang, A. Bayen, and Yi Wu. The surprising effectiveness of mappo in cooperative, multi-agent games. ArXiv, abs/2103.01955, 2021.
> - [Foerster et. al., 2018] Jakob Foerster, Gregory Farquhar, Triantafyllos Afouras, Nantas Nardelli, and Shimon Whiteson. Counterfactual multi-agent policy gradients. In Proceedings of the AAAI Conference on Artificial Intelligence, volume 32, 2018.

---

> > ### Author Response · Authors · 2021-08-26
> > **Response to Reviewer n8hf**
> >
> > Dear Reviewer n8hf:
> >
> > We sincerely hope that your questions regarding the technical details of our paper have been addressed. We are more than happy to answer if anything further your feel unclear or confused.
> >
> > Authors.

---

### Decision · Program_Chairs · 2021-09-27

**Decision:**

Accept (Poster)

**Comment:**

The paper studies how to reduce the variance of stochastic gradient estimates in policy gradient approaches to Multi-agent Reinforcement Learning (MARL). Focusing on the popular Centralized Training Decentralized Execution setting, the authors provide a theoretical analysis that quantifies excess variance and connects it to the number of agents and agents' local advantages. They build on these insights by empirically showing how these insights affect COMA, and by deriving a new optimal baseline (OB) which is empirically validated in StarCraft and multi-agent MUJOCO environments.

Reviewers were positive about the importance of the problem addressed in this work. They largely agreed that the question of policy gradient variance has not been widely studied in the MARL community, but has important implications and potential for impact. The paper was considered clear and well written.

At the same time, several concerns were raised. In particular, the assumptions of the analysis are idealised, and findings do not always translate into corresponding empirical gains.

During the rebuttal and discussion phase, reviewers were generally satisfied with the clarifications provided by the authors and recommend acceptance. Unfortunately, the most negative reviewer did not engage beyond the initial review. Therefore, the AC examined the concerns raised in the initial review and judged these as sufficiently addressed by the authors.

The paper is assessed as meeting the bar for acceptance, based on the important theoretical insights and high potential for impact on the MARL community. The authors are strongly encouraged to take on board all reviewer comments to further improve clarity and potential for impact in the camera ready version.